# Improved Distributed Principal Component Analysis

**Maria-Florina Balcan**
School of Computer Science
Carnegie Mellon University
ninamf@cs.cmu.edu

**Vandana Kanchanapally**
School of Computer Science
Georgia Institute of Technology
vvandana@gatech.edu

**Yingyu Liang**
Department of Computer Science
Princeton University
yingyul@cs.princeton.edu

**David Woodruff**
Almaden Research Center
IBM Research
dpwoodru@us.ibm.com

## Abstract

We study the distributed computing setting in which there are multiple servers, each holding a set of points, who wish to compute functions on the union of their point sets. A key task in this setting is Principal Component Analysis (PCA), in which the servers would like to compute a low dimensional subspace capturing as much of the variance of the union of their point sets as possible. Given a procedure for approximate PCA, one can use it to approximately solve problems such as $k$-means clustering and low rank approximation. The essential properties of an approximate distributed PCA algorithm are its communication cost and computational efficiency for a given desired accuracy in downstream applications. We give new algorithms and analyses for distributed PCA which lead to improved communication and computational costs for $k$-means clustering and related problems. Our empirical study on real world data shows a speedup of orders of magnitude, preserving communication with only a negligible degradation in solution quality. Some of these techniques we develop, such as a general transformation from a constant success probability subspace embedding to a high success probability subspace embedding with a dimension and sparsity independent of the success probability, may be of independent interest.

## 1 Introduction

Since data is often partitioned across multiple servers [20, 7, 18], there is an increased interest in computing on it in the distributed model. A basic tool for distributed data analysis is Principal Component Analysis (PCA). The goal of PCA is to find an $r$-dimensional (affine) subspace that captures as much of the variance of the data as possible. Hence, it can reveal low-dimensional structure in very high dimensional data. Moreover, it can serve as a preprocessing step to reduce the data dimension in various machine learning tasks, such as Non-Negative Matrix Factorization (NNMF) [15] and Latent Dirichlet Allocation (LDA) [3].

In the distributed model, approximate PCA was used by Feldman et al. [9] for solving a number of shape fitting problems such as $k$-means clustering, where the approximation is in the form of a *coreset*, and has the property that local coresets can be easily combined across servers into a global coreset, thereby providing an approximate PCA to the union of the data sets. Designing small coresets therefore leads to communication-efficient protocols. Coresets have the nice property that their size typically does not depend on the number $n$ of points being approximated. A beautiful property of the coresets developed in [9] is that for approximate PCA their size also only depends linearly on the dimension $d$, whereas previous coresets depended quadratically on $d$ [8]. This gives the best known communication protocols for approximate PCA and $k$-means clustering.

Despite this recent exciting progress, several important questions remain. First, can we improve the communication further as a function of the number of servers, the approximation error, and other parameters of the downstream applications (such as the number $k$ of clusters in $k$-means clustering)? Second, while preserving optimal or nearly-optimal communication, can we improve the computational costs of the protocols? We note that in the protocols of Feldman et al. each server has to run a singular value decomposition (SVD) on her local data set, while additional work needs to be performed to combine the outputs of each server into a global approximate PCA. Third, are these algorithms practical and do they scale well with large-scale datasets? In this paper we give answers to the above questions. To state our results more precisely, we first define the model and the problems.

**Communication Model.** In the distributed setting, we consider a set of $s$ nodes $\mathcal{V} = \{v_i, 1 \le i \le s\}$, each of which can communicate with a central coordinator $v_0$. On each node $v_i$, there is a local data matrix $\mathbf{P}_i \in \mathbb{R}^{n_i \times d}$ having $n_i$ data points in $d$ dimension ($n_i > d$). The global data $\mathbf{P} \in \mathbb{R}^{n \times d}$ is then a concatenation of the local data matrix, i.e. $\mathbf{P}^\top = [\mathbf{P}_1^\top, \mathbf{P}_2^\top, \dots, \mathbf{P}_s^\top]$ and $n = \sum_{i=1}^{s} n_i$. Let $p_i$ denote the $i$-th row of $\mathbf{P}$. Throughout the paper, we assume that the data points are centered to have zero mean, i.e., $\sum_{i=1}^{n} p_i = 0$. Uncentered data requires a rank-one modification to the algorithms, whose communication and computation costs are dominated by those in the other steps.

**Approximate PCA and $\ell_2$-Error Fitting.** For a matrix $\mathbf{A} = [a_{ij}]$, let $\|\mathbf{A}\|_F^2 = \sum_{i,j} a_{ij}^2$ be its Frobenius norm, and let $\sigma_i(\mathbf{A})$ be the $i$-th singular value of $\mathbf{A}$. Let $\mathbf{A}^{(t)}$ denote the matrix that contains the first $t$ columns of $\mathbf{A}$. Let $L_{\mathbf{X}}$ denote the linear subspace spanned by the columns of $\mathbf{X}$. For a point $p$, let $\pi_L(p)$ be its projection onto subspace $L$ and let $\pi_{\mathbf{X}}(p)$ be shorthand for $\pi_{L_{\mathbf{X}}}(p)$. For a point $p \in \mathbb{R}^d$ and a subspace $L \subseteq \mathbb{R}^d$, we denote the squared distance between $p$ and $L$ by

$$d^2(p, L) := \min_{q \in L} \|p - q\|_2^2 = \|p - \pi_L(p)\|_2^2.$$

**Definition 1.** The linear (or affine) $r$-Subspace $k$-Clustering on $\mathbf{P} \in \mathbb{R}^{n \times d}$ is

$$\min_{\mathcal{L}} d^2(\mathbf{P}, \mathcal{L}) := \sum_{i=1}^{n} \min_{L \in \mathcal{L}} d^2(p_i, L) \tag{1}$$

where $\mathbf{P}$ is an $n \times d$ matrix whose rows are $p_1, \dots, p_n$, and $\mathcal{L} = \{L_j\}_{j=1}^{k}$ is a set of $k$ centers, each of which is an $r$-dimensional linear (or affine) subspace.

PCA is a special case when $k = 1$ and the center is an $r$-dimensional subspace. This optimal $r$-dimensional subspace is spanned by the top $r$ right singular vectors of $\mathbf{P}$, also known as the principal components, and can be found using the singular value decomposition (SVD). Another special case of the above is $k$-means clustering when the centers are points ($r = 0$). Constrained versions of this problem include NNMF where the $r$-dimensional subspace should be spanned by positive vectors, and LDA which assumes a prior distribution defining a probability for each $r$-dimensional subspace. We will primarily be concerned with relative-error approximation algorithms, for which we would like to output a set $\mathcal{L}'$ of $k$ centers for which $d^2(\mathbf{P}, \mathcal{L}') \le (1 + \epsilon) \min_{\mathcal{L}} d^2(\mathbf{P}, \mathcal{L})$.

For approximate distributed PCA, the following protocol is implicit in [9]: each server $i$ computes its top $O(r/\epsilon)$ principal components $\mathbf{Y}_i$ of $\mathbf{P}_i$ and sends them to the coordinator. The coordinator stacks the $O(r/\epsilon) \times d$ matrices $\mathbf{Y}_i$ on top of each other, forming an $O(sr/\epsilon) \times d$ matrix $\mathbf{Y}$, and computes the top $r$ principal components of $\mathbf{Y}$, and returns these to the servers. This provides a relative-error approximation to the PCA problem. We refer to this algorithm as Algorithm disPCA.

**Our Contributions.** Our results are summarized as follows.

*Improved Communication:* We improve the communication cost for using distributed PCA for $k$-means clustering and similar $\ell_2$-fitting problems. The best previous approach is to use Corollary 4.5 in [9], which shows that given a data matrix $\mathbf{P}$, if we project the rows onto the space spanned by the top $O(k/\epsilon^2)$ principal components, and solve the $k$-means problem in this subspace, we obtain a $(1+\epsilon)$-approximation. In the distributed setting, this would require first running Algorithm disPCA with parameter $r = O(k/\epsilon^2)$, and thus communication at least $O(skd/\epsilon^3)$ to compute the $O(k/\epsilon^2)$ global principal components. Then one can solve a distributed $k$-means problem in this subspace, and an $\alpha$-approximation in it translates to an overall $\alpha(1 + \epsilon)$ approximation.

Our Theorem 3 shows that it suffices to run Algorithm disPCA while only incurring $O(skd/\epsilon^2)$ communication to compute the $O(k/\epsilon^2)$ global principal components, preserving the $k$-means solution cost up to a $(1 + \epsilon)$-factor. Our communication is thus a $1/\epsilon$ factor better, and illustrates that

for downstream applications it is sometimes important to "open up the box" rather than to directly use the guarantees of a generic PCA algorithm (which would give $O(skd/\epsilon^3)$ communication). One feature of this approach is that by using the distributed $k$-means algorithm in [2] on the projected data, the coordinator can sample points from the servers proportional to their local $k$-means cost solutions, which reduces the communication roughly by a factor of $s$, which would come from each server sending their local $k$-means coreset to the coordinator. Furthermore, before applying the above approach, one can first run any other dimension reduction to dimension $d'$ so that the $k$-means cost is preserved up to certain accuracy. For example, if we want a $1+\epsilon$ approximation factor, we can set $d' = O(\log n/\epsilon^2)$ by a Johnson-Lindenstrauss transform; if we want a larger $2+\epsilon$ approximation factor, we can set $d' = O(k/\epsilon^2)$ using [4]. In this way the parameter $d$ in the above communication cost bound can be replaced by $d'$. Note that unlike these dimension reductions, our algorithm for projecting onto principal components is deterministic and does not incur error probability.

*Improved Computation:* We turn to the computational cost of Algorithm disPCA, which to the best of our knowledge has not been addressed. A major bottleneck is that each player is computing a singular value decomposition (SVD) of its point set $\mathbf{P}_i$, which takes $\min(n_i d^2, n_i^2 d)$ time. We change Algorithm disPCA to instead have each server first sample an oblivious subspace embedding (OSE) [22, 5, 19, 17] matrix $\mathbf{H}_i$, and instead run the algorithm on the point set defined by the rows of $\mathbf{H}_i\mathbf{P}_i$. Using known OSEs, one can choose $\mathbf{H}_i$ to have only a single non-zero entry per column and thus $\mathbf{H}_i\mathbf{P}_i$ can be computed in $\mathrm{nnz}(\mathbf{P}_i)$ time. Moreover, the number of rows of $\mathbf{H}_i$ is $O(d^2/\epsilon^2)$, which may be significantly less than the original $n_i$ number of rows. This number of rows can be further reduced to $O(d\log^{O(1)} d/\epsilon^2)$ if one is willing to spend $O(\mathrm{nnz}(\mathbf{P}_i)\log^{O(1)} d/\epsilon)$ time [19]. We note that the number of non-zero entries of $\mathbf{H}_i\mathbf{P}_i$ is no more than that of $\mathbf{P}_i$.

One technical issue is that each of $s$ servers is locally performing a subspace embedding, which succeeds with only constant probability. If we want a single non-zero entry per column of $\mathbf{H}_i$, to achieve success probability $1 - O(1/s)$ so that we can union bound over all $s$ servers succeeding, we naively would need to increase the number of rows of $\mathbf{H}_i$ by a factor linear in $s$. We give a general technique, which takes a subspace embedding that succeeds with constant probability as a black box, and show how to perform a procedure which applies it $O(\log 1/\delta)$ times independently and from these applications finds one which is guaranteed to succeed with probability $1 - \delta$. Thus, in this setting the players can compute a subspace embedding of their data in $\mathrm{nnz}(\mathbf{P}_i)$ time, for which the number of non-zero entries of $\mathbf{H}_i\mathbf{P}_i$ is no larger than that of $\mathbf{P}_i$, and without incurring this additional factor of $s$. This may be of independent interest.

It may still be expensive to perform the SVD of $\mathbf{H}_i\mathbf{P}_i$ and for the coordinator to perform an SVD on $\mathbf{Y}$ in Algorithm disPCA. We therefore replace the SVD computation with a randomized approximate SVD computation with spectral norm error. Our contribution here is to analyze the error in distributed PCA and $k$-means after performing these speedups.

*Empirical Results:* Our speedups result in significant computational savings. The randomized techniques we use reduce the time by orders of magnitude on medium and large-scal data sets, while preserving the communication cost. Although the theory predicts a new small additive error because of our speedups, in our experiments the solution quality was only negligibly affected.

**Related Work** A number of algorithms for approximate distributed PCA have been proposed [21, 14, 16, 9], but either without theoretical guarantees, or without considering communication. Most closely related to our work is [9, 12]. [9] observes the top singular vectors of the local data is its summary and the union of these summaries is a summary of the global data, i.e., Algorithm disPCA. [12] studies algorithms in the arbitrary partition model in which each server holds a matrix $\mathbf{P}_i$ and $\mathbf{P} = \sum_{i=1}^{s} \mathbf{P}_i$. More details and more related work can be found in the appendix.

## 2 Tradeoff between Communication and Solution Quality

Algorithm disPCA for distributed PCA is suggested in [21, 9], which consists of a local stage and a global stage. In the local stage, each node performs SVD on its local data matrix, and communicates the first $t_1$ singular values $\mathbf{\Sigma}_i^{(t_1)}$ and the first $t_1$ right singular vectors $\mathbf{V}_i^{(t_1)}$ to the central coordinator. Then in the global stage, the coordinator concatenates $\mathbf{\Sigma}_i^{(t_1)}(\mathbf{V}_i^{(t_1)})^\top$ to form a matrix $\mathbf{Y}$, and performs SVD on it to get the first $t_2$ right singular vectors.

To get some intuition, consider the easy case when the data points actually lie in an $r$-dimensional subspace. We can run Algorithm disPCA with $t_1 = t_2 = r$. Since $\mathbf{P}_i$ has rank $r$, its projection to

$$\mathbf{P} = \begin{bmatrix} \mathbf{P}_1 \\ \vdots \\ \mathbf{P}_s \end{bmatrix} \xrightarrow[\text{Local PCA}]{\text{Local PCA}} \begin{bmatrix} \mathbf{\Sigma}_1^{(t_1)} \left( \mathbf{V}_1^{(t_1)} \right)^\top \\ \vdots \\ \mathbf{\Sigma}_s^{(t_1)} \left( \mathbf{V}_s^{(t_1)} \right)^\top \end{bmatrix} = \begin{bmatrix} \mathbf{Y}_1 \\ \vdots \\ \mathbf{Y}_s \end{bmatrix} = \mathbf{Y} \xrightarrow{\text{Global PCA}} \mathbf{V}^{(t_2)}$$

Figure 1: The key points of the algorithm disPCA.

the subspace spanned by its first $t_1 = r$ right singular vectors, $\widehat{\mathbf{P}}_i = \mathbf{U}_i \mathbf{\Sigma}_i^{(r)} (\mathbf{V}_i^{(r)})^\top$, is identical to $\mathbf{P}_i$. Then we only need to do PCA on $\widehat{\mathbf{P}}$, the concatenation of $\widehat{\mathbf{P}}_i$. Observing that $\widehat{\mathbf{P}} = \widetilde{\mathbf{U}} \mathbf{Y}$ where $\widetilde{\mathbf{U}}$ is orthonormal, it suffices to compute SVD on $\mathbf{Y}$, and only $\mathbf{\Sigma}_i^{(r)} \mathbf{V}_i^{(r)}$ needs to be communicated. In the general case when the data may have rank higher than $r$, it turns out that one needs to set $t_1$ sufficiently large, so that $\widehat{\mathbf{P}}_i$ approximates $\mathbf{P}_i$ well enough and does not introduce too much error into the final solution. In particular, the following *close projection* property about SVD is useful:

**Lemma 1.** *Suppose $\mathbf{A}$ has SVD $\mathbf{A} = \mathbf{U}\mathbf{\Sigma}\mathbf{V}$ and let $\widehat{\mathbf{A}} = \mathbf{A}\mathbf{V}^{(t)}(\mathbf{V}^{(t)})^\top$ denote its SVD truncation. If $t = O(r/\epsilon)$, then for any $d \times r$ matrix $\mathbf{X}$ with orthonormal columns,*

$$0 \le \|\mathbf{A}\mathbf{X} - \widehat{\mathbf{A}}\mathbf{X}\|_F^2 \le \epsilon d^2(\mathbf{A}, L_{\mathbf{X}}), \quad \text{and} \quad 0 \le \|\mathbf{A}\mathbf{X}\|_F^2 - \|\widehat{\mathbf{A}}\mathbf{X}\|_F^2 \le \epsilon d^2(\mathbf{A}, L_{\mathbf{X}}).$$

This means that the projections of $\widehat{\mathbf{A}}$ and $\mathbf{A}$ on any $r$-dimensional subspace are close, when the projected dimension $t$ is sufficiently large compared to $r$. Now, note that the difference between $\|\mathbf{P} - \mathbf{P}\mathbf{X}\mathbf{X}^\top\|_F^2$ and $\|\widehat{\mathbf{P}} - \widehat{\mathbf{P}}\mathbf{X}\mathbf{X}^\top\|_F^2$ is only related to $\|\mathbf{P}\mathbf{X}\|_F^2 - \|\widehat{\mathbf{P}}\mathbf{X}\|_F^2 = \sum_i[\|\mathbf{P}_i\mathbf{X}\|_F^2 - \|\widehat{\mathbf{P}}_i\mathbf{X}\|_F^2]$. Each term in which is bounded by the lemma. So we can use $\widehat{\mathbf{P}}$ as a proxy for $\mathbf{P}$ in the PCA task. Again, computing PCA on $\widehat{\mathbf{P}}$ is equivalent to computing SVD on $\mathbf{Y}$, as done in Algorithm disPCA. These lead to the following theorem, which is implicit in [9], stating that the algorithm can produce a $(1 + \epsilon)$-approximation for the distributed PCA problem.

**Theorem 2.** *Suppose Algorithm disPCA takes parameters $t_1 \ge r + \lceil 4r/\epsilon \rceil - 1$ and $t_2 = r$. Then*

$$\|\mathbf{P} - \mathbf{P}\mathbf{V}^{(r)}(\mathbf{V}^{(r)})^\top\|_F^2 \le (1 + \epsilon) \min_{\mathbf{X}} \|\mathbf{P} - \mathbf{P}\mathbf{X}\mathbf{X}^\top\|_F^2$$

*where the minimization is over $d \times r$ orthonormal matrices $\mathbf{X}$. The communication is $O(\frac{srd}{\epsilon})$ words.*

### 2.1 Guarantees for Distributed $\ell_2$-Error Fitting

Algorithm disPCA can also be used as a pre-processing step for applications such as $\ell_2$-error fitting. In this section, we prove the correctness of Algorithm disPCA as pre-processing for these applications. In particular, we show that by setting $t_1, t_2$ sufficiently large, the objective value of any solution merely changes when the original data $\mathbf{P}$ is replaced the projected data $\tilde{\mathbf{P}} = \mathbf{P}\mathbf{V}^{(t_2)}(\mathbf{V}^{(t_2)})^\top$. Therefore, the projected data serves as a proxy of the original data, i.e., any distributed algorithm can be applied on the projected data to get a solution on the original data. As the dimension is lower, the communication cost is reduced. Formally,

**Theorem 3.** *Let $t_1 = t_2 = O(rk/\epsilon^2)$ in Algorithm disPCA for $\epsilon \in (0, 1/3)$. Then there exists a constant $c_0 \ge 0$ such that for any set of $k$ centers $\mathcal{L}$ in $r$-Subspace $k$-Clustering,*

$$(1 - \epsilon)d^2(\mathbf{P}, \mathcal{L}) \le d^2(\tilde{\mathbf{P}}, \mathcal{L}) + c_0 \le (1 + \epsilon)d^2(\mathbf{P}, \mathcal{L}).$$

The theorem implies that any $\alpha$-approximate solution $\mathcal{L}$ on the projected data $\tilde{\mathbf{P}}$ is a $(1 + 3\epsilon)\alpha$-approximation on the original data $\mathbf{P}$. To see this, let $\mathcal{L}^*$ denote the optimal solution. Then

$$(1 - \epsilon)d^2(\mathbf{P}, \mathcal{L}) \le d^2(\tilde{\mathbf{P}}, \mathcal{L}) + c_0 \le \alpha d^2(\tilde{\mathbf{P}}, \mathcal{L}^*) + c_0 \le \alpha(1 + \epsilon)d^2(\mathbf{P}, \mathcal{L}^*)$$

which leads to $d^2(\mathbf{P}, \mathcal{L}) \le (1 + 3\epsilon)\alpha d^2(\mathbf{P}, \mathcal{L}^*)$. In other words, the distributed PCA step only introduces a small multiplicative approximation factor of $(1 + 3\epsilon)$.

The key to prove the theorem is the close projection property of the algorithm (Lemma 4): for any low dimensional subspace spanned by $\mathbf{X}$, the projections of $\mathbf{P}$ and $\tilde{\mathbf{P}}$ on the subspace are close. In

---

**Algorithm 1** Distributed $k$-means clustering

---

**Input:** $\{\mathbf{P}_i\}_{i=1}^s$, $k \in \mathbb{N}_+$ and $\epsilon \in (0, 1/2)$, a non-distributed $\alpha$-approximation algorithm $\mathcal{A}_\alpha$
  1: Run Algorithm disPCA with $t_1 = t_2 = O(k/\epsilon^2)$ to get $\mathbf{V}$, and send $\mathbf{V}$ to all nodes.
  2: Run the distributed $k$-means clustering algorithm in [2] on $\{\mathbf{P}_i \mathbf{V}\mathbf{V}^\top\}_{i=1}^s$, using $\mathcal{A}_\alpha$ as a subroutine, to get $k$ centers $\mathcal{L}$.
**Output:** $\mathcal{L}$.

---

particular, we choose $\mathbf{X}$ to be the orthonormal basis of the subspace spanning the centers. Then the difference between the objective values of $\mathbf{P}$ and $\tilde{\mathbf{P}}$ can be decomposed into two terms depending only on $\|\mathbf{PX} - \tilde{\mathbf{P}}\mathbf{X}\|_F^2$ and $\|\mathbf{PX}\|_F^2 - \|\tilde{\mathbf{P}}\mathbf{X}\|_F^2$ respectively, which are small as shown by the lemma. The complete proof of Theorem 3 is provided in the appendix.

**Lemma 4.** *Let $t_1 = t_2 = O(k/\epsilon)$ in Algorithm* disPCA. *Then for any $d \times k$ matrix $\mathbf{X}$ with orthonormal columns, $0 \le \|\mathbf{PX} - \tilde{\mathbf{P}}\mathbf{X}\|_F^2 \le \epsilon d^2(\mathbf{P}, L_\mathbf{X})$, and $0 \le \|\mathbf{PX}\|_F^2 - \|\tilde{\mathbf{P}}\mathbf{X}\|_F^2 \le \epsilon d^2(\mathbf{P}, L_\mathbf{X})$.*

**Proof Sketch:** We first introduce some auxiliary variables for the analysis, which act as intermediate connections between $\mathbf{P}$ and $\tilde{\mathbf{P}}$. Imagine we perform two kinds of projections: first project $\mathbf{P}_i$ to $\widehat{\mathbf{P}}_i = \mathbf{P}_i \mathbf{V}_i^{(t_1)} (\mathbf{V}_i^{(t_1)})^\top$, then project $\widehat{\mathbf{P}}_i$ to $\overline{\mathbf{P}}_i = \widehat{\mathbf{P}}_i \mathbf{V}^{(t_2)} (\mathbf{V}^{(t_2)})^\top$. Let $\widehat{\mathbf{P}}$ denote the vertical concatenation of $\widehat{\mathbf{P}}_i$ and let $\overline{\mathbf{P}}$ denote the vertical concatenation of $\overline{\mathbf{P}}_i$. These variables are designed so that the difference between $\mathbf{P}$ and $\widehat{\mathbf{P}}$ and that between $\widehat{\mathbf{P}}$ and $\overline{\mathbf{P}}$ are easily bounded.

Our proof then proceeds by first bounding these differences, and then bounding that between $\overline{\mathbf{P}}$ and $\tilde{\mathbf{P}}$. In the following we sketch the proof for the second statement, while the other statement can be proved by a similar argument. See the appendix for details.

$$\|\mathbf{PX}\|_F^2 - \|\tilde{\mathbf{P}}\mathbf{X}\|_F^2 \;=\; \left[ \|\mathbf{PX}\|_F^2 - \|\widehat{\mathbf{P}}\mathbf{X}\|_F^2 \right] + \left[ \|\widehat{\mathbf{P}}\mathbf{X}\|_F^2 - \|\overline{\mathbf{P}}\mathbf{X}\|_F^2 \right] + \left[ \|\overline{\mathbf{P}}\mathbf{X}\|_F^2 - \|\tilde{\mathbf{P}}\mathbf{X}\|_F^2 \right].$$

The first term is just $\sum_{i=1}^s \left[ \|\mathbf{P}_i\mathbf{X}\|_F^2 - \|\widehat{\mathbf{P}}_i\mathbf{X}\|_F^2 \right]$, each of which can be bounded by Lemma 1, since $\widehat{\mathbf{P}}_i$ is the SVD truncation of $\mathbf{P}$. The second term can be bounded similarly. The more difficult part is the third term. Note that $\overline{\mathbf{P}}_i = \widehat{\mathbf{P}}_i \mathbf{Z}$, $\tilde{\mathbf{P}}_i = \mathbf{P}_i \mathbf{Z}$ where $\mathbf{Z} := \mathbf{V}^{(t_2)} (\mathbf{V}^{(t_2)})^\top \mathbf{X}$, leading to $\|\overline{\mathbf{P}}\mathbf{X}\|_F^2 - \|\tilde{\mathbf{P}}\mathbf{X}\|_F^2 = \sum_{i=1}^s \left[ \|\widehat{\mathbf{P}}_i\mathbf{Z}\|_F^2 - \|\mathbf{P}_i\mathbf{Z}\|_F^2 \right]$. Although $\mathbf{Z}$ is not orthonormal as required by Lemma 1, we prove a generalization (Lemma 7 in the appendix) which can be applied to show that the third term is indeed small. □

**Application to $k$-Means Clustering** To see the implication, consider the $k$-means clustering problem. We can first perform any other possible dimension reduction to dimension $d'$ so that the $k$-means cost is preserved up to accuracy $\epsilon$, and then run Algorithm disPCA and finally run any distributed $k$-means clustering algorithm on the projected data to get a good approximate solution. For example, in the first step we can set $d' = O(\log n/\epsilon^2)$ using a Johnson-Lindenstrauss transform, or we can perform no reduction and simply use the original data.

As a concrete example, we can use original data ($d' = d$), then run Algorithm disPCA, and finally run the distributed clustering algorithm in [2] which uses any non-distributed $\alpha$-approximation algorithm as a subroutine and computes a $(1 + \epsilon)\alpha$-approximate solution. The resulting algorithm is presented in Algorithm 1.

**Theorem 5.** *With probability at least $1 - \delta$, Algorithm 1 outputs a $(1 + \epsilon)^2 \alpha$-approximate solution for distributed $k$-means clustering. The total communication cost of Algorithm 1 is $O(\frac{sk}{\epsilon^2})$ vectors in $\mathbb{R}^d$ plus $O\left( \frac{1}{\epsilon^4}(\frac{k^2}{\epsilon^2} + \log \frac{1}{\delta}) + sk \log \frac{sk}{\delta} \right)$ vectors in $\mathbb{R}^{O(k/\epsilon^2)}$.*

## 3   Fast Distributed PCA

**Subspace Embeddings** One can significantly improve the time of the distributed PCA algorithms by using subspace embeddings, while keeping similar guarantees as in Lemma 4, which suffice for $l_2$-error fitting. More precisely, a subspace embedding matrix $\mathbf{H} \in \mathbb{R}^{\ell \times n}$ for a matrix $\mathbf{A} \in \mathbb{R}^{n \times d}$ has the property that for all vectors $y \in \mathbb{R}^d$, $\|\mathbf{H}\mathbf{A}y\|_2 = (1 \pm \epsilon)\|\mathbf{A}y\|_2$. Suppose independently,

each node $v_i$ chooses a random subspace embedding matrix $\mathbf{H}_i$ for its local data $\mathbf{P}_i$. Then, they run Algorithm disPCA on the embedded data $\{\mathbf{H}_i\mathbf{P}_i\}_{i=1}^s$ instead of on the original data $\{\mathbf{P}_i\}_{i=1}^s$.

The work of [22] pioneered subspace embeddings. The recent fast sparse subspace embeddings [5] and its optimizations [17, 19] are particularly suitable for large scale sparse data sets, since their running time is linear in the number of non-zero entries in the data matrix, and they also preserve the sparsity of the data. The algorithm takes as input an $n \times d$ matrix $\mathbf{A}$ and a parameter $\ell$, and outputs an $\ell \times d$ embedded matrix $\mathbf{A}' = \mathbf{H}\mathbf{A}$ (the embedded matrix $\mathbf{H}$ does need to be built explicitly). The embedded matrix is constructed as follows: initialize $\mathbf{A}' = \mathbf{0}$; for each row in $\mathbf{A}$, multiply it by $+1$ or $-1$ with equal probability, then add it to a row in $\mathbf{A}'$ chosen uniformly at random.

The success probability is constant, while we need to set it to be $1 - \delta$ where $\delta = \Theta(1/s)$. Known results which preserve the number of non-zero entries of $\mathbf{H}$ to be 1 per column increase the dimension of $\mathbf{H}$ by a factor of $s$. To avoid this, we propose an approach to boost the success probability by computing $O(\log \frac{1}{\delta})$ independent embeddings, each with only constant success probability, and then run a cross validation style procedure to find one which succeeds with probability $1 - \delta$. More precisely, we compute the SVD of all embedded matrices $\mathbf{H}_j\mathbf{A} = \mathbf{U}_j\mathbf{\Sigma}_j\mathbf{V}_j^\top$, and find a $j \in [r]$ such that for at least half of the indices $j' \neq j$, all singular values of $\mathbf{\Sigma}_j\mathbf{V}_j^\top\mathbf{V}_{j'}\mathbf{\Sigma}_{j'}^\top$ are in $[1 \pm O(\epsilon)]$ (see Algorithm 4 in the appendix). The reason why such an embedding $\mathbf{H}_j\mathbf{A}$ succeeds with high probability is as follows. Any two successful embeddings $\mathbf{H}_j\mathbf{A}$ and $\mathbf{H}_{j'}\mathbf{A}$, by definition, satisfy that $\|\mathbf{H}_j\mathbf{A}x\|_2^2 = (1 \pm O(\epsilon))\|\mathbf{H}_{j'}\mathbf{A}x\|_2^2$ for all $x$, which we show is equivalent to passing the test on the singular values. Since with probability at least $1 - \delta$, $9/10$ fraction of the embeddings are successful, it follows that the one we choose is successful with probability $1 - \delta$.

**Randomized SVD** The exact SVD of an $n \times d$ matrix is impractical in the case when $n$ or $d$ is large. Here we show that the randomized SVD algorithm from [11] can be applied to speed up the computation without compromising the quality of the solution much. We need to use their specific form of randomized SVD since the error is with respect to the spectral norm, rather than the Frobenius norm, and so can be much smaller as needed by our applications.

The algorithm first probes the row space of the $\ell \times d$ input matrix $\mathbf{A}$ with an $\ell \times 2t$ random matrix $\mathbf{\Omega}$ and orthogonalizes the image of $\mathbf{\Omega}$ to get a basis $\mathbf{Q}$ (i.e., QR-factorize $\mathbf{A}^\top\mathbf{\Omega}$); projects the data to this basis and computes the SVD factorization on the smaller matrix $\mathbf{A}\mathbf{Q}$. It also performs $q$ power iterations to push the basis towards the top $t$ singular vectors.

**Fast Distributed PCA for $l_2$-Error Fitting** We modify Algorithm disPCA by first having each node do a subspace embedding locally, then replace each SVD invocation with a randomized SVD invocation. We thus arrive at Algorithm 2. For $\ell_2$-error fitting problems, by combining approximation guarantees of the randomized techniques with that of distributed PCA, we are able to prove:

**Theorem 6.** *Suppose Algorithm 2 takes $\epsilon \in (0, 1/2]$, $t_1 = t_2 = O(\max\{\frac{k}{\epsilon^2}, \log \frac{s}{\delta}\}), \ell = O(\frac{d^2}{\epsilon^2}), q = O(\max\{\log \frac{d}{\epsilon}, \log \frac{sk}{\epsilon}\})$ as input, and sets the failure probability of each local subspace embedding to $\delta' = \delta/2s$. Let $\tilde{\mathbf{P}} = \mathbf{P}\mathbf{V}\mathbf{V}^\top$. Then with probability at least $1 - \delta$, there exists a constant $c_0 \geq 0$, such that for any set of $k$ points $\mathcal{L}$,*

$$(1 - \epsilon)d^2(\mathbf{P}, \mathcal{L}) - \epsilon\|\mathbf{P}\mathbf{X}\|_F^2 \leq d^2(\tilde{\mathbf{P}}, \mathcal{L}) + c_0 \leq (1 + \epsilon)d^2(\mathbf{P}, \mathcal{L}) + \epsilon\|\mathbf{P}\mathbf{X}\|_F^2$$

*where $\mathbf{X}$ is an orthonormal matrix whose columns span $\mathcal{L}$. The total communication is $O(skd/\epsilon^2)$ and the total time is $O\left(\mathrm{nnz}(\mathbf{P}) + s\left[\frac{d^3k}{\epsilon^4} + \frac{k^2d^2}{\epsilon^6}\right]\log\frac{d}{\epsilon}\log\frac{sk}{\delta\epsilon}\right)$.*

**Proof Sketch:** It suffices to show that $\tilde{\mathbf{P}}$ enjoys the close projection property as in Lemma 4, i.e., $\|\mathbf{P}\mathbf{X} - \tilde{\mathbf{P}}\mathbf{X}\|_F^2 \approx 0$ and $\|\mathbf{P}\mathbf{X}\|_F^2 - \|\tilde{\mathbf{P}}\mathbf{X}\|_F^2 \approx 0$ for any orthonormal matrix whose columns span a low dimensional subspace. Note that Algorithm 2 is just running Algorithm disPCA (with randomized SVD) on $\mathbf{T}\mathbf{P}$ where $\mathbf{T} = \mathrm{diag}(\mathbf{H}_1, \mathbf{H}_2, \ldots, \mathbf{H}_s)$, so we first show that $\mathbf{T}\tilde{\mathbf{P}}$ enjoys this property. But now exact SVD is replaced with randomized SVD, for which we need to use the spectral error bound to argue that the error introduced is small. More precisely, for a matrix $\mathbf{A}$ and its SVD truncation $\widehat{\mathbf{A}}$ computed by randomized SVD, it is guaranteed that the spectral norm of $\mathbf{A} - \widehat{\mathbf{A}}$ is small, then $\|(\mathbf{A} - \widehat{\mathbf{A}})\mathbf{X}\|_F$ is small for any $\mathbf{X}$ with small Frobenius norm, in particular, the orthonormal basis spanning a low dimensional subspace. This then suffices to guarantee $\mathbf{T}\tilde{\mathbf{P}}$ enjoys the close projection property. Given this, it suffices to show that $\tilde{\mathbf{P}}$ enjoys this property as $\mathbf{T}\tilde{\mathbf{P}}$, which follows from the definition of a subspace embedding. $\square$

---
**Algorithm 2** Fast Distributed PCA for $l_2$-Error Fitting
---
**Input:** $\{\mathbf{P}_i\}_{i=1}^s$; parameters $t_1, t_2$ for Algorithm disPCA; $\ell, q$ for randomized techniques.
  1: **for** each node $v_i \in \mathcal{V}$ **do**
  2:     Compute subspace embedding $\mathbf{P}'_i = \mathbf{H}_i \mathbf{P}_i$.
  3: **end for**
  4: Run Algorithm disPCA on $\{\mathbf{P}'_i\}_{i=1}^s$ to get $\mathbf{V}$, where the SVD is randomized.
**Output:** $\mathbf{V}$.
---

## 4 Experiments

Our focus is to show the randomized techniques used in Algorithm 2 reduce the time taken significantly without compromising the quality of the solution. We perform experiments for three tasks: rank-$r$ approximation, $k$-means clustering and principal component regression (PCR).

**Datasets** We choose the following real world datasets from UCI repository [1] for our experiments. For low rank approximation and $k$-means clustering, we choose two medium size datasets News-Groups ($18774 \times 61188$) and MNIST ($70000 \times 784$), and two large-scale Bag-of-Words datasets: NYTimes news articles (BOWnytimes) ($300000 \times 102660$) and PubMed abstracts (BOWpubmed) ($8200000 \times 141043$). We use $r = 10$ for rank-$r$ approximation and $k = 10$ for $k$-means clustering. For PCR, we use MNIST and further choose YearPredictionMSD ($515345 \times 90$), CTslices ($53500 \times 386$), and a large dataset MNIST8m ($800000 \times 784$).

**Experimental Methodology** The algorithms are evaluated on a star network. The number of nodes is $s = 25$ for medium-size datasets, and $s = 100$ for the larger ones. We distribute the data over the nodes using a weighted partition, where each point is distributed to the nodes with probability proportional to the node's weight chosen from the power law with parameter $\alpha = 2$.

For each projection dimension, we first construct the projected data using distributed PCA. For low rank approximation, we report the ratio between the cost of the obtained solution to that of the solution computed by SVD on the global data. For $k$-means, we run the algorithm in [2] (with Lloyd's method as a subroutine) on the projected data to get a solution. Then we report the ratio between the cost of the above solution to that of a solution obtained by running Lloyd's method directly on the global data. For PCR, we perform regression on the projected data to get a solution. Then we report the ratio between the error of the above solution to that of a solution obtained by PCR directly on the global data. We stop the algorihtm if it takes more than 24 hours. For each projection dimension and each algorithm with randomness, the average ratio over 5 runs is reported.

**Results** Figure 2 shows the results for low rank approximation. We observe that the error of the fast distributed PCA is comparable to that of the exact solution computed directly on the global data. This is also observed for distributed PCA with one or none of subspace embedding and randomized SVD. Furthermore, the error of the fast PCA is comparable to that of normal PCA, which means that the speedup techniques merely affects the accuracy of the solution. The second row shows the computational time, which suggests a significant decrease in the time taken to run the fast distributed PCA. For example, on NewsGroups, the time of the fast distributed PCA improves over that of normal distributed PCA by a factor between 10 to 100. On the large dataset BOWpubmed, the normal PCA takes too long to finish and no results are presented, while the speedup versions produce good results in reasonable time. The use of the randomized techniques gives us a good performance improvement while keeping the solution quality almost the same.

Figure 3 and Figure 4 show the results for $k$-means clustering and PCR respectively. Similar to that for low rank approximation, we observe that the distributed solutions are almost as good as that computed directly on the global data, and the speedup merely affects the solution quality. We again observe a huge decrease in the running time by the speedup techniques.

**Acknowledgments** This work was supported in part by NSF grants CCF-0953192, CCF-1451177, CCF-1101283, and CCF-1422910, ONR grant N00014-09-1-0751, and AFOSR grant FA9550-09-1-0538. David Woodruff would like to acknowledge the XDATA program of the Defense Advanced Research Projects Agency (DARPA), administered through Air Force Research Laboratory contract FA8750-12-C0323, for supporting this work.

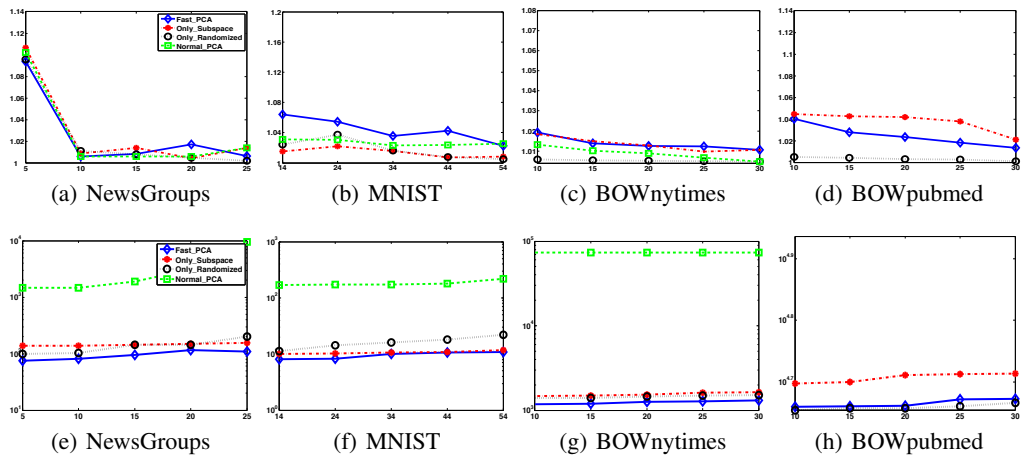

Figure 2: Low rank approximation. First row: error (normalized by baseline) v.s. projection dimension. Second row: time v.s. projection dimension.

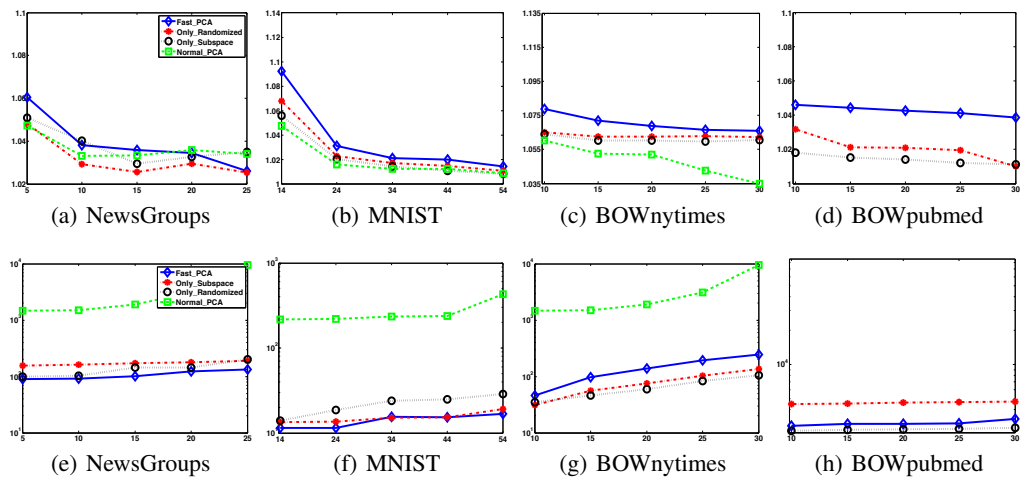

Figure 3: $k$-means clustering. First row: cost (normalized by baseline) v.s. projection dimension. Second row: time v.s. projection dimension.

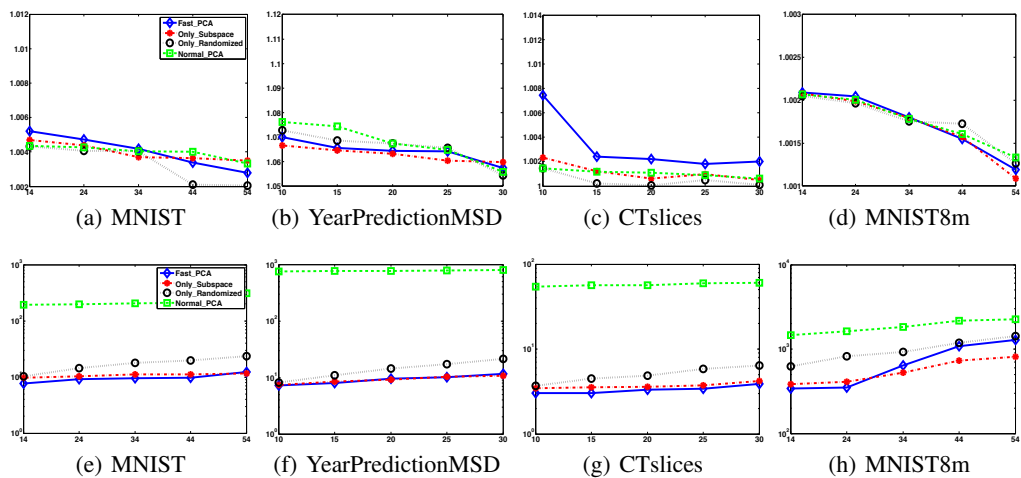

Figure 4: PCR. First row: error (normalized by baseline) v.s. projection dimension. Second row: time v.s. projection dimension.

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
