[Supplementary Material]

## A   Related Work

A number of algorithms for approximate distributed PCA have been proposed [21, 14, 16, 9], but either without theoretical guarantees, or without considering communication. [21] proposed an algorithm but provided no analysis on the tradeoff between communication and approximation. Most closely related to our work is [9], which observes that the top singular vectors of the local point set can be viewed as its summary and the union of the local summaries can be viewed as a summary of the global data, i.e., Algorithm disPCA discussed above.

In [12] the authors study algorithms in the arbitrary partition model in which each server holds a matrix $\mathbf{P}_i$ and $\mathbf{P} = \sum_{i=1}^{s} \mathbf{P}_i$. Thus, each row of $\mathbf{P}$ is additively shared across the $s$ servers, whereas in our model each row of $\mathbf{P}$ belongs to a single server, though duplicate rows are allowed. Our model is motivated by applications in which points are indecomposable entities. As our model is a special case of the arbitrary partition model, we can achieve more efficient algorithms. For instance, our distributed PCA algorithms provide much stronger guarantees, see, e.g., Lemma 4, which are needed for the downstream $k$-means application. Moreover, our $k$-means algorithms are more general, in the sense that they do not make a well-separability assumption, and more efficient in that the communication of [12] is $O(sd^2) + s(k/\epsilon)^{O(1)}$ words as opposed to our $O(sdk/\epsilon^2) + sk + (k/\epsilon)^{O(1)}$.

After the announce of this work, [6] improve the guarantee for the $k$-means application in two ways. First, they tighten the result in [9], showing that projecting to just the $O(k/\epsilon)$ rather than $O(k/\epsilon^2)$ top singular vectors is sufficient to approximate $k$-means with $(1 + \epsilon)$ error. Second, they show that performing a Johnson-Lindenstrauss transformation down to $O(k/\epsilon^2)$ dimension gives $(1 + \epsilon)$ approximation without requiring a $\log(n)$ dependence. This can be used as a preprocessing step before our algorithm, replacing $d$ with $O(k/\epsilon^2)$ in our communication bounds. They further show how to reduce the dimension to $O(k/\epsilon)$ using only $O(sk/\epsilon)$vectors, but by a technique different from distributed PCA.

Other related work includes the recent [10] (see also the references therein), who give a deterministic streaming algorithm for low rank approximation in which each point of $\mathbf{P}$ is seen one at a time and uses $O(dk/\epsilon)$ words of communication. Their algorithm naturally gives an $O(sdk/\epsilon)$ communication algorithm for low rank approximation in the distributed model. However, their algorithm for PCA doesn't satisfy the stronger guarantees of Lemma 4, and therefore it is unclear how to use it for $k$-means clustering. It also involves an SVD computation for each point, making the overall computation per server $O(n_i dr^2/\epsilon^2)$, which is slower than what we achieve, and it is not clear how their algorithm can exploit sparsity.

Speeding up large scale PCA using different versions of subspace embeddings was also considered in [13], though not in a distributed setting and not for $\ell_2$-error shape fitting problems. Also, their error guarantees are in terms of the $r$-th singular value gap, and are incomparable to ours.

## B   Guarantees for Distributed PCA

### B.1   Proof of Lemma 1

We first prove a generalization of Lemma 1.

**Lemma 7.** *Let* $\mathbf{A} \in \mathbb{R}^{n \times d}$ *be an* $n \times d$ *matrix with singular value decomposition* $\mathbf{A} = \mathbf{U}\mathbf{\Sigma}\mathbf{V}^\top$. *Let* $\epsilon \in (0, 1]$ *and* $r, t \in \mathbb{N}_+$ *with* $d - 1 \geq t \geq r + \lceil r/\epsilon \rceil - 1$, *and let* $\widehat{\mathbf{A}} = \mathbf{A}\mathbf{V}^{(t)}(\mathbf{V}^{(t)})^\top$. *Then for any matrix* $\mathbf{X}$ *with* $d$ *rows and* $\|\mathbf{X}\|_F^2 \leq r$, *we have*

$$\|(\mathbf{A} - \widehat{\mathbf{A}})\mathbf{X}\|_F^2 = \|\mathbf{A}\mathbf{X}\|_F^2 - \|\widehat{\mathbf{A}}\mathbf{X}\|_F^2 \leq \epsilon \sum_{i=r+1}^{d} \sigma_i^2(\mathbf{A}).$$

*Proof.* The proof follows the idea in the proof of Lemma 6.1 in [9].

For convenience, let $\overline{\boldsymbol{\Sigma}^{(t)}}$ denote the diagonal matrix that contains the first $t$ diagonal entries in $\boldsymbol{\Sigma}$ and is 0 otherwise. Then $\widehat{\mathbf{A}} = \mathbf{U}\overline{\boldsymbol{\Sigma}^{(t)}}\mathbf{V}^\top$ We first have

$$
\begin{aligned}
\|\mathbf{A}\mathbf{X}\|_F^2 - \|\widehat{\mathbf{A}}\mathbf{X}\|_F^2 &= \|\mathbf{U}\boldsymbol{\Sigma}\mathbf{V}^\top\mathbf{X}\|_F^2 - \|\mathbf{U}\overline{\boldsymbol{\Sigma}^{(t)}}\mathbf{V}^\top\mathbf{X}\|_F^2 \\
&= \|\boldsymbol{\Sigma}\mathbf{V}^\top\mathbf{X}\|_F^2 - \|\overline{\boldsymbol{\Sigma}^{(t)}}\mathbf{V}^\top\mathbf{X}\|_F^2 \\
&= \|(\boldsymbol{\Sigma} - \overline{\boldsymbol{\Sigma}^{(t)}})\mathbf{V}^\top\mathbf{X}\|_F^2 \\
&= \|\mathbf{U}(\boldsymbol{\Sigma} - \overline{\boldsymbol{\Sigma}^{(t)}})\mathbf{V}^\top\mathbf{X}\|_F^2 \\
&= \|\mathbf{A}\mathbf{X} - \widehat{\mathbf{A}}\mathbf{X}\|_F^2.
\end{aligned}
$$

where the second and fourth equalities follow since $\mathbf{U}$ has orthonormal columns, and the third equality follows since for $\mathbf{M} = \mathbf{V}^\top\mathbf{X}$ we have

$$
\begin{aligned}
\|\boldsymbol{\Sigma}\mathbf{M}\|_F^2 - \|\overline{\boldsymbol{\Sigma}^{(t)}}\mathbf{M}\|_F^2 &= \sum_{i=1}^{d}\sum_{j=1}^{d}\sigma_i^2(\mathbf{A})m_{ij}^2 - \sum_{i=1}^{t}\sum_{j=1}^{d}\sigma_i^2(\mathbf{A})m_{ij}^2 \\
&= \sum_{i=t+1}^{d}\sum_{j=1}^{d}\sigma_i^2(\mathbf{A})m_{ij}^2 = \|(\boldsymbol{\Sigma} - \overline{\boldsymbol{\Sigma}^{(t)}})\mathbf{M}\|_F^2.
\end{aligned}
$$

Next, we bound $\|\mathbf{A}\mathbf{X} - \widehat{\mathbf{A}}\mathbf{X}\|_F^2$. We have

$$
\|\mathbf{A}\mathbf{X} - \widehat{\mathbf{A}}\mathbf{X}\|_F^2 = \|(\boldsymbol{\Sigma} - \overline{\boldsymbol{\Sigma}^{(t)}})\mathbf{V}^\top\mathbf{X}\|_F^2 \leq \|(\boldsymbol{\Sigma} - \overline{\boldsymbol{\Sigma}^{(t)}})\|_S^2\|\mathbf{X}\|_F^2 = r\sigma_{t+1}^2(\mathbf{A})
$$

where the inequality follows because the spectral norm is consistent with the Euclidean norm. This implies the lemma since

$$
r\sigma_{t+1}^2(\mathbf{A}) \leq \epsilon(t - r + 1)\sigma_{t+1}^2(\mathbf{A}) \leq \epsilon\sum_{i=r+1}^{t+1}\sigma_i^2(\mathbf{A}) \leq \epsilon\sum_{i=r+1}^{d}\sigma_i^2(\mathbf{A}). \tag{2}
$$

where the first inequality follows for our choice of $t$. $\qquad\square$

Then Lemma 1 immediately follows from Lemma 7 since any $d \times r$ orthonormal matrix $\mathbf{A}$ has $\|\mathbf{A}\|_F^2 \leq r$, and $\sum_{i=r+1}^{d}\sigma_i^2(\mathbf{A}) \leq d^2(\mathbf{A}, L_\mathbf{X})$ by the property of the singular value decomposition.

## B.2 Proof of Theorem 2

**Theorem 2.** *Suppose Algorithm* disPCA *takes parameters* $t_1 \geq r + \lceil 4r/\epsilon\rceil - 1$ *and* $t_2 = r$, *and outputs* $\mathbf{V}^{(r)}$. *Then*

$$
\|\mathbf{P} - \mathbf{P}\mathbf{V}^{(r)}(\mathbf{V}^{(r)})^\top\|_F^2 \leq (1 + \epsilon)\min_{\mathbf{X}} d^2(\mathbf{P}, L_\mathbf{X})
$$

*where the minimization is over* $d \times r$ *orthonormal matrices* $\mathbf{X}$. *The communication is* $O(\frac{srd}{\epsilon})$ *words.*

*Proof.* Recall the notations: $\widehat{\mathbf{P}}_i := \mathbf{P}_i\mathbf{V}_i^{(t_1)}(\mathbf{V}_i^{(t_1)})^\top$ is the data obtained by applying local PCA on local data $P_i$, and $\widehat{\mathbf{P}}$ is the concatenation of $\widehat{\mathbf{P}}_i$. Now let $\mathbf{X}^*$ denote the optimal subspace for $\mathbf{P}$. Our goal is to show that the distance between $\mathbf{P}$ and the subspace spanned by $\mathbf{V}^{(r)}$ is close to that between $\mathbf{P}$ and the subspace spanned by $\mathbf{X}^*$.

To get some intuition, see Figure 5 for an illustration. We let $a$ denote the distance between $\mathbf{P}$ and $L_{\mathbf{V}^{(r)}}$, that is, $a := d^2(\mathbf{P}, L_{\mathbf{V}^{(r)}}) = \|\mathbf{P} - \mathbf{P}\mathbf{V}^{(r)}(\mathbf{V}^{(r)})^\top\|_F^2$. Similarly, let $b$ denote the distance between $\mathbf{P}$ and $L_{\mathbf{X}^*}$, $c$ denote that between $\widehat{\mathbf{P}}$ and $L_{\mathbf{V}^{(r)}}$, $d$ denote that between $\widehat{\mathbf{P}}$ and $L_{\mathbf{X}^*}$. Then our goal is to show $a - b$ is small. Since

$$
a - b = (a - c) + (c - d) + (d - b),
$$

it suffices to bound each of the three terms on the right hand side.

Figure 5: Illustration for the proof of Theorem 2.

First, we note that the optimal principal components for $\widehat{\mathbf{P}}$ are $\mathbf{V}^{(r)}$, so $c - d \leq 0$. This is because $\widehat{\mathbf{P}} = \tilde{\mathbf{U}}\mathbf{Y}$ where $\tilde{\mathbf{U}}$ is a block-diagonal matrix with blocks $\mathbf{U}_1, \ldots, \mathbf{U}_s$, and thus the right singular vectors of $\mathbf{Y}$ are also the right singular vectors of $\widehat{\mathbf{P}}$.

Now, what is left is to bound $(a - c)$ and $(d - b)$. They are differences between the distances from $\mathbf{P}$ and $\widehat{\mathbf{P}}$ to some low dimensional subspace, for which Lemma 1 is useful. Formally, we have the following claim.

**Claim 1.** *For any orthonormal matrix $\mathbf{X}$ of size $d \times r$,*

$$d^2(\widehat{\mathbf{P}}, L_{\mathbf{X}}) - d^2(\mathbf{P}, L_{\mathbf{X}}) = \Delta(\mathbf{X}) - c_0$$

*where $\Delta(\mathbf{X}) := \|\mathbf{P}\mathbf{X}\|_F^2 - \|\widehat{\mathbf{P}}\mathbf{X}\|_F^2$ and $c_0 := \|\mathbf{P}\|_F^2 - \|\widehat{\mathbf{P}}\|_F^2$. Furthermore,*

$$0 \leq \Delta(\mathbf{X}) \leq \epsilon d^2(\mathbf{P}, L_{\mathbf{X}}), \quad c_0 \geq 0.$$

*Proof.* By Pythagorean Theorem,

$$d^2(\widehat{\mathbf{P}}, L_{\mathbf{X}}) - d^2(\mathbf{P}, L_{\mathbf{X}}) = (\|\widehat{\mathbf{P}}\|_F^2 - \|\widehat{\mathbf{P}}\mathbf{X}\|_F^2) - (\|\mathbf{P}\|_F^2 - \|\mathbf{P}\mathbf{X}\|_F^2) = \Delta(\mathbf{X}) - c_0.$$

The bound on $\Delta(\mathbf{X})$ follows from the fact that

$$\Delta(\mathbf{X}) = \|\mathbf{P}\mathbf{X}\|_F^2 - \|\widehat{\mathbf{P}}\mathbf{X}\|_F^2 = \sum_i [\|\mathbf{P}_i\mathbf{X}\|_F^2 - \|\widehat{\mathbf{P}}_i\mathbf{X}\|_F^2]$$

and apply Lemma 1 on each term. The bound on $c_0$ follows from Pythagorean Theorem. $\quad\square$

Applying this claim, we have $a - c = c_0 - \Delta(\mathbf{V}^{(r)})$ and $d - b = \Delta(\mathbf{V}^*) - c_0$, and

$$(a - c) + (d - b) = \Delta(\mathbf{V}^*) - \Delta(\mathbf{V}^{(r)}) \leq \epsilon d^2(\mathbf{P}, L_{\mathbf{X}^*}).$$

This completes the proof. $\quad\square$

**Note** A refinement of the proof of Lemma 1 leads to the following data dependent bound.

**Lemma 8.** *The statement in Lemma 7 holds if $t > \tau(\mathbf{A}, r, \epsilon)$ where*

$$\tau(\mathbf{A}, r, \epsilon) := \underset{t}{\operatorname{argmin}} \left\{ \sigma_t^2(\mathbf{A}) \leq \frac{\epsilon}{r} \sum_{i > r} \sigma_i^2(\mathbf{A}) \right\}.$$

*Furthermore, $\tau(\mathbf{A}, r, \epsilon) = O(\frac{r}{\epsilon})$.*

*Proof.* Note that the bound on $t$ is only used in proving (2), for which $t > \tau(\mathbf{A}, r, \epsilon)$ suffices. $\tau(\mathbf{A}, r, \epsilon) = O(\frac{r}{\epsilon})$ follows by definition. $\quad\square$

**Theorem 9.** *Suppose Algorithm* disPCA *takes parameters* $t_1 \geq \max_i \tau(\mathbf{P}_i, r, \epsilon)$ *and* $t_2 = r$, *and outputs* $\mathbf{V}^{(r)}$. *Then*

$$\|\mathbf{P} - \mathbf{P}\mathbf{V}^{(r)}(\mathbf{V}^{(r)})^\top\|_F^2 \leq (1 + \epsilon) \min_{\mathbf{X}} d^2(\mathbf{P}, L_{\mathbf{X}})$$

*where the minimization is over orthonormal matrices* $\mathbf{X} \in \mathbb{R}^{d \times r}$. *The total communication cost is* $O(sd \max_i \tau(\mathbf{P}_i, r, \epsilon))$ *words.*

$\tau(\mathbf{P}_i, r, \epsilon)$ is typically much less than $O(r/\epsilon)$ in practice. This provides an explanation for the fact that $t_1$ much smaller than $O(r/\epsilon)$ can still lead to good solution for many practical instances. Similar data dependent bounds can be derived for the other theorems in our paper.

# C Guarantees for Distributed $\ell_2$-Error Fitting

## C.1 Proof of Lemma 4

Recall that $\tilde{\mathbf{P}}_i$ denotes the projection of the original data $\mathbf{P}_i$ to $\mathbf{V}^{(t)}$, and $\tilde{\mathbf{P}}$ denotes their concatenation. We further introduce some intermediate variables for our analysis. Imagine we perform two projections: first project $\mathbf{P}_i$ to $\widehat{\mathbf{P}}_i = \mathbf{P}_i \mathbf{V}_i^{(t)}(\mathbf{V}_i^{(t)})^\top$, then project $\widehat{\mathbf{P}}_i$ to $\overline{\mathbf{P}}_i = \widehat{\mathbf{P}}_i \mathbf{V}^{(t)}(\mathbf{V}^{(t)})^\top$ where $t = t_1 = t_2$. Let $\widehat{\mathbf{P}}$ denote the vertical concatenation of $\widehat{\mathbf{P}}_i$ and let $\overline{\mathbf{P}}$ denote the vertical concatenation of $\overline{\mathbf{P}}_i$, i.e.

$$\widehat{\mathbf{P}} = \begin{bmatrix} \widehat{\mathbf{P}}_1 \\ \vdots \\ \widehat{\mathbf{P}}_s \end{bmatrix} \quad \text{and} \quad \overline{\mathbf{P}} = \begin{bmatrix} \overline{\mathbf{P}}_1 \\ \vdots \\ \overline{\mathbf{P}}_s \end{bmatrix}$$

**Lemma 4.** *Let* $t_1 = t_2 \geq k + \lceil 8k/\epsilon \rceil - 1$ *in Algorithm* disPCA *for* $k \in \mathbb{N}_+$ *and* $\epsilon \in (0, 1)$. *Then for any* $d \times k$ *matrix* $\mathbf{X}$ *with orthonormal columns,*

$$0 \leq \quad \|\mathbf{P}\mathbf{X} - \tilde{\mathbf{P}}\mathbf{X}\|_F^2 \quad \leq \epsilon d^2(\mathbf{P}, L_{\mathbf{X}}), \tag{3}$$

$$0 \leq \quad \|\mathbf{P}\mathbf{X}\|_F^2 - \|\tilde{\mathbf{P}}\mathbf{X}\|_F^2 \quad \leq \epsilon d^2(\mathbf{P}, L_{\mathbf{X}}). \tag{4}$$

*Proof.* Before going to the proof, we note that unlike in Lemma 7, $\|\mathbf{P}\mathbf{X} - \tilde{\mathbf{P}}\mathbf{X}\|_F^2$ may not equal $\|\mathbf{P}\mathbf{X}\|_F^2 - \|\tilde{\mathbf{P}}\mathbf{X}\|_F^2$ since multiple SVD are applied.

For the first statement (3), we have

$$\|\mathbf{P}\mathbf{X} - \tilde{\mathbf{P}}\mathbf{X}\|_F^2 \quad \leq \quad 2\|\mathbf{P}\mathbf{X} - \widehat{\mathbf{P}}\mathbf{X}\|_F^2 \tag{5}$$

$$+ \quad 2\|\widehat{\mathbf{P}}\mathbf{X} - \overline{\mathbf{P}}\mathbf{X}\|_F^2 \tag{6}$$

$$+ \quad 2\|\overline{\mathbf{P}}\mathbf{X} - \tilde{\mathbf{P}}\mathbf{X}\|_F^2. \tag{7}$$

For (5), we have by Lemma 7

$$\|\mathbf{P}\mathbf{X} - \widehat{\mathbf{P}}\mathbf{X}\|_F^2 = \sum_{i=1}^{s} \|\mathbf{P}_i\mathbf{X} - \widehat{\mathbf{P}}_i\mathbf{X}\|_F^2 \leq \sum_{i=1}^{s} \frac{\epsilon}{4} d^2(\mathbf{P}_i, L_{\mathbf{X}}) = \frac{\epsilon}{8} d^2(\mathbf{P}, L_{\mathbf{X}}). \tag{8}$$

Similarly, for (6) we have by Lemma 7

$$\|\widehat{\mathbf{P}}\mathbf{X} - \overline{\mathbf{P}}\mathbf{X}\|_F^2 \leq \frac{\epsilon}{8} d^2(\widehat{\mathbf{P}}, L_{\mathbf{X}}). \tag{9}$$

To bound (7), let $\mathbf{Y} = \mathbf{V}^{(t)}(\mathbf{V}^{(t)})^\top \mathbf{X}$. Then by definition, $\overline{\mathbf{P}}_i \mathbf{X} = \widehat{\mathbf{P}}_i \mathbf{Y}$ and $\tilde{\mathbf{P}}_i \mathbf{X} = \mathbf{P}_i \mathbf{Y}$. By Lemma 7, we have

$$\|\overline{\mathbf{P}}\mathbf{X} - \tilde{\mathbf{P}}\mathbf{X}\|_F^2 \quad = \quad \sum_{i=1}^{s} \|\widehat{\mathbf{P}}_i \mathbf{Y} - \mathbf{P}_i \mathbf{Y}\|_F^2 \tag{10}$$

$$\leq \quad \sum_{i=1}^{s} \frac{\epsilon}{8} \sum_{i=r+1}^{s} \sigma_i^2(\mathbf{P}_i) \leq \frac{\epsilon}{8} \sum_{i=1}^{s} d^2(\mathbf{P}_i, L_{\mathbf{X}}) = \frac{\epsilon}{8} d^2(\mathbf{P}, L_{\mathbf{X}}). \tag{11}$$

Combining (8)(9) and (11) leads to

$$\|\mathbf{PX} - \tilde{\mathbf{P}}\mathbf{X}\|_F^2 \le \frac{\epsilon}{2}d^2(\mathbf{P}, L_\mathbf{X}) + \frac{\epsilon}{4}d^2(\widehat{\mathbf{P}}, L_\mathbf{X}). \tag{12}$$

We now only need to bound $d^2(\widehat{\mathbf{P}}, L_\mathbf{X})$ is similar to $d^2(\mathbf{P}, L_\mathbf{X})$, which is done in Claim 1. The first statement then follows.

For the second statement (4), we have a similar argument.

$$\begin{aligned}
\|\mathbf{PX}\|_F^2 - \|\tilde{\mathbf{P}}\mathbf{X}\|_F^2 &= \|\mathbf{PX}\|_F^2 - \|\widehat{\mathbf{P}}\mathbf{X}\|_F^2 \tag{13}\\
&+ \|\widehat{\mathbf{P}}\mathbf{X}\|_F^2 - \|\overline{\mathbf{P}}\mathbf{X}\|_F^2 \tag{14}\\
&+ \|\overline{\mathbf{P}}\mathbf{X}\|_F^2 - \|\tilde{\mathbf{P}}\mathbf{X}\|_F^2. \tag{15}
\end{aligned}$$

For (13), we have by Lemma 7

$$\|\mathbf{PX}\|_F^2 - \|\widehat{\mathbf{P}}\mathbf{X}\|_F^2 = \sum_{i=1}^{s} \left[\|\mathbf{P}_i\mathbf{X}\|_F^2 - \|\widehat{\mathbf{P}}_i\mathbf{X}\|_F^2\right] \le \sum_{i=1}^{s} \frac{\epsilon}{4}d^2(\mathbf{P}_i, L_\mathbf{X}) = \frac{\epsilon}{4}d^2(\mathbf{P}, L_\mathbf{X}). \tag{16}$$

Similarly, for (14) we have by Lemma 7

$$\|\widehat{\mathbf{P}}\mathbf{X}\|_F^2 - \|\overline{\mathbf{P}}\mathbf{X}\|_F^2 \le \frac{\epsilon}{4}d^2(\widehat{\mathbf{P}}, L_\mathbf{X}). \tag{17}$$

By Lemma 7, we have

$$\begin{aligned}
\|\overline{\mathbf{P}}\mathbf{X}\|_F^2 - \|\tilde{\mathbf{P}}\mathbf{X}\|_F^2 &= \sum_{i=1}^{s} \left[\|\widehat{\mathbf{P}}_i\mathbf{Y}\|_F^2 - \|\mathbf{P}_i\mathbf{Y}\|_F^2\right]\\
&\le \sum_{i=1}^{s} \frac{\epsilon}{4} \sum_{i=r+1}^{s} \sigma_i^2(\mathbf{P}_i) \le \frac{\epsilon}{4} \sum_{i=1}^{s} d^2(\mathbf{P}_i, L_\mathbf{X}) = \frac{\epsilon}{4}d^2(\mathbf{P}, L_\mathbf{X}). \tag{18}
\end{aligned}$$

Combining (16)(17) and (18) leads to

$$\|\mathbf{PX}\|_F^2 - \|\tilde{\mathbf{P}}\mathbf{X}\|_F^2 \le \frac{\epsilon}{2}d^2(\mathbf{P}, L_\mathbf{X}) + \frac{\epsilon}{4}d^2(\widehat{\mathbf{P}}, L_\mathbf{X}). \tag{19}$$

The second statement then follows from (19) and Claim 1. $\square$

## C.2 Proof of Theorem 3

The following weak triangle inequality is useful for our analysis.

**Fact 1.** *For any $a, b \in \mathbb{R}$ and $\epsilon \in (0, 1)$, $|a^2 - b^2| \le \frac{3(a-b)^2}{\epsilon} + 2\epsilon a^2$.*

*Proof.* Either $|a| \le \frac{|a-b|}{\epsilon}$ or $|a - b| \le \epsilon|a|$, so we have $|a||a - b| \le \frac{(a-b)^2}{\epsilon} + \epsilon a^2$. This leads to

$$|a^2 - b^2| = |a - b||a + b| \le |a - b|(|2a| + |b - a|) = 2|a||a - b| + (a - b)^2 \le \frac{2(a - b)^2}{\epsilon} + 2\epsilon a^2 + (a - b)^2$$

which completes the proof. $\square$

We first prove the theorem for the special case of $k$-means clustering, and the same argument leads to the guarantee for general $l_2$-error fitting problems. Note that because we use the weak triangle inequality, we lose a factor of $1/\epsilon$. Thus, we require $t_1 = t_2 = O(k/\epsilon^2)$, instead of $O(k/\epsilon)$ as in Lemma 4.

**Theorem 10.** *Let $t_1 = t_2 \ge k + \lceil 4k/\epsilon^2 \rceil - 1$ in Algorithm* disPCA.*Then there exists a constant $c_0 \ge 0$, such that for any set of $k$ points $\mathcal{L}$,*

$$(1 - \epsilon)d^2(\mathbf{P}, \mathcal{L}) \le d^2(\tilde{\mathbf{P}}, \mathcal{L}) + c_0 \le (1 + \epsilon)d^2(\mathbf{P}, \mathcal{L}).$$

Figure 6: Illustration for the proof of Theorem 10.

*Proof.* The proof follows that in [9], with slight modification for the distributed setting.

Let $\mathbf{X} \in \mathbb{R}^{d \times k}$ has orthonormal columns that span $\mathcal{L}$; see Figure 6 for an illustration. Then the costs of $\mathbf{P}$ and $\tilde{\mathbf{P}}$ can be decomposed into two parts: one part is from $\mathbf{P}$ (or $\tilde{\mathbf{P}}$) to its projection on $L_{\mathbf{X}}$, and the other part is from the projection to the centers. Then we can compare the two parts separately.

Let $\tilde{p}_i$ be the point in $\tilde{\mathbf{P}}$ corresponding to $p_i$ in $\mathbf{P}$. Let $c_0 = \|\mathbf{P}\|_F^2 - \|\tilde{\mathbf{P}}\|_F^2$. Then by Pythagorean theorem we have

$$|d^2(\mathbf{P}, \mathcal{L}) - d^2(\tilde{\mathbf{P}}, \mathcal{L}) - c_0| \leq \left| d^2(\mathbf{P}, L_{\mathbf{X}}) - d^2(\tilde{\mathbf{P}}, L_{\mathbf{X}}) - c_0 \right| + \left| \sum_{i=1}^{|\mathbf{P}|} \left[ d(\pi_{\mathbf{X}}(p_i), \mathcal{L})^2 - d(\pi_{\mathbf{X}}(\tilde{p}_i), \mathcal{L})^2 \right] \right|.$$

For the first part, we have by Pythagorean theorem

$$d^2(\mathbf{P}, L_{\mathbf{X}}) - d^2(\tilde{\mathbf{P}}, L_{\mathbf{X}}) - c_0 = (\|\mathbf{P}\|_F^2 - \|\mathbf{P}\mathbf{X}\|_F^2) - (\|\tilde{\mathbf{P}}\|_F^2 - \|\tilde{\mathbf{P}}\mathbf{X}\|_F^2) - c_0 = \|\tilde{\mathbf{P}}\mathbf{X}\|_F^2 - \|\mathbf{P}\mathbf{X}\|_F^2. \quad (20)$$

For the second part, by Fact 1 we have

$$
\begin{aligned}
\sum_{i=1}^{|\mathbf{P}|} \left| d(\pi_{\mathbf{X}}(p_i), \mathcal{L})^2 - d(\pi_{\mathbf{X}}(\tilde{p}_i), \mathcal{L})^2 \right| &\leq \sum_{i=1}^{|\mathbf{P}|} \left[ \frac{12 d(\pi_{\mathbf{X}}(p_i), \pi_{\mathbf{X}}(\tilde{p}_i))^2}{\epsilon} + \frac{\epsilon}{2} d(\pi_{\mathbf{X}}(p_i), \mathcal{L})^2 \right] \\
&= \frac{12}{\epsilon} \|(\mathbf{P} - \tilde{\mathbf{P}})\mathbf{X}\|_F^2 + \frac{\epsilon}{2} \sum_{i=1}^{|\mathbf{P}|} d(\pi_{\mathbf{X}}(p_i), \mathcal{L})^2 \\
&\leq \frac{12}{\epsilon} \|(\mathbf{P} - \tilde{\mathbf{P}})\mathbf{X}\|_F^2 + \frac{\epsilon}{2} \sum_{i=1}^{|\mathbf{P}|} d(p_i, \mathcal{L})^2. \quad (21)
\end{aligned}
$$

We first note that $d^2(\mathbf{P}, L_{\mathbf{X}}) \leq d^2(\mathbf{P}, \mathcal{L})$. For the other terms in (20)(21), we need to use Lemma 4 with accuracy $\epsilon^2$ (instead of $\epsilon$). This then leads to the theorem. $\square$

The general statement for $\ell_2$-error geometric fitting problems follows from the same argument.

**Theorem 3.** *Let $t_1 = t_2 = O(rk/\epsilon^2)$ in Algorithm* disPCA *for $\epsilon \in (0, 1/3)$. Then there exists a constant $c_0 \geq 0$ such that for any set of $k$ centers $\mathcal{L}$ in $r$-Subspace $k$-Clustering,*

$$(1 - \epsilon)d^2(\mathbf{P}, \mathcal{L}) \leq d^2(\tilde{\mathbf{P}}, \mathcal{L}) + c_0 \leq (1 + \epsilon)d^2(\mathbf{P}, \mathcal{L}).$$

# D    Fast Distributed PCA

## D.1    Proofs for Subspace Embedding

---

**Algorithm 3** Fast Sparse Subspace Embedding [5]

---

**Input:** parameters $n, \ell \in \mathbb{N}_+$.
 1: Let $h : [n] \mapsto [\ell]$ be a random map, so that for each $i \in [n], h(i) = j$ for $j \in [\ell]$ with probability $1/\ell$.
 2: Let $\mathbf{\Phi}$ be an $\ell \times n$ binary matrix with $\mathbf{\Phi}_{h(i),i} = 1$, and all remaining entries 0.
 3: Let $\mathbf{\Sigma}$ be an $n \times n$ diagonal matrix, with each diagonal entry independently chosen as $+1$ or $-1$ with equal probability.
**Output:** $\mathbf{H} = \mathbf{\Phi}\mathbf{\Sigma}$.

---

The construction of the embedding matrix $\mathbf{H}$ is presented in Algorithm 3. Note that the embedding matrix $\mathbf{H}$ does not need to be built explicitly; we can compute the embedding $\mathbf{HA}$ for an given matrix $\mathbf{A}$ in a direct and faster way. Algorithm 3 has the following guarantee.

**Theorem 11.** *[5, 17, 19] Suppose $n > d$ and $\ell = O(\frac{d^2}{\epsilon^2})$. With probability at least 99/100, $\|\mathbf{HA}y\|_2 = (1 \pm \epsilon)\|\mathbf{A}y\|_2$ for all vectors $y \in \mathbb{R}^d$. Moreover, $\mathbf{HA}$ can be computed in time $O(\text{nnz}(\mathbf{A}))$ where $\text{nnz}(\mathbf{A})$ is the number of non-zero entries in $\mathbf{A}$.*

**Lemma 12.** *Let $\epsilon \in (0, 1/2]$ and $k, t \in \mathbb{N}_+$ with $d - 1 \geq t \geq k + \lceil 4k/\epsilon \rceil - 1$. Suppose Algorithm disPCA takes input $\{\mathbf{H}_i \mathbf{P}_i\}_{i=1}^s$ and outputs $\mathbf{V}^{(t)}$. Let $\tilde{\mathbf{P}} = \mathbf{P}\mathbf{V}^{(t)}(\mathbf{V}^{(t)})^\top$. Then for any $d \times k$ matrix $\mathbf{X}$ with orthonormal columns,*

$$\|\mathbf{PX} - \tilde{\mathbf{P}}\mathbf{X}\|_F^2 \leq \epsilon d^2(\mathbf{P}, L_\mathbf{X}),$$
$$\left|\|\mathbf{PX}\|_F^2 - \|\tilde{\mathbf{P}}\mathbf{X}\|_F^2\right| \leq 3\epsilon\|\mathbf{PX}\|_F^2 + \epsilon d^2(\mathbf{P}, L_\mathbf{X}).$$

*Proof.* First note that the input to Algorithm disPCA is $\mathbf{TP}$ where $\mathbf{T}$ is a block-diagonal matrix with blocks $\mathbf{H}_1, \ldots, \mathbf{H}_s$. Then the projection of the input to $\mathbf{V}^{(t)}$ is $\mathbf{TP}\mathbf{V}^{(t)}(\mathbf{V}^{(t)})^\top = \mathbf{T}\tilde{\mathbf{P}}$. By Lemma 4, for any $d \times k$ matrix $\mathbf{X}$ with orthonormal columns, we have

$$0 \leq \quad \|\mathbf{TPX} - \mathbf{T}\tilde{\mathbf{P}}\mathbf{X}\|_F^2 \quad \leq \frac{\epsilon}{4}d^2(\mathbf{TP}, L_\mathbf{X}), \tag{22}$$

$$0 \leq \|\mathbf{TPX}\|_F^2 - \|\mathbf{T}\tilde{\mathbf{P}}\mathbf{X}\|_F^2 \quad \leq \frac{\epsilon}{4}d^2(\mathbf{TP}, L_\mathbf{X}). \tag{23}$$

By properties of $\mathbf{T}$, we have

$$\|\mathbf{TPX} - \mathbf{T}\tilde{\mathbf{P}}\mathbf{X}\|_F^2 = \|\mathbf{T}(\mathbf{PX} - \tilde{\mathbf{P}}\mathbf{X})\|_F^2 \geq (1 - \epsilon)\|\mathbf{PX} - \tilde{\mathbf{P}}\mathbf{X}\|_F^2$$

and

$$d^2(\mathbf{TP}, L_\mathbf{X}) = \|\mathbf{TP} - \mathbf{TPXX}^\top\|_F^2 \leq (1 + \epsilon)\|\mathbf{P} - \mathbf{PXX}^\top\|_F^2 = (1 + \epsilon)d^2(\mathbf{P}, L_\mathbf{X}).$$

Combined with (22), these lead to the first claim.

Similarly, we also have $\|\mathbf{TPX}\|_F^2 = (1 \pm \epsilon)\|\mathbf{PX}\|_F^2$ and $\|\mathbf{T}\tilde{\mathbf{P}}\mathbf{X}\|_F^2 = (1 \pm \epsilon)\|\tilde{\mathbf{P}}\mathbf{X}\|_F^2$. Plugging these into (23), we obtain

$$-3\epsilon\|\mathbf{PX}\|_F^2 \leq \|\mathbf{PX}\|_F^2 - \|\tilde{\mathbf{P}}\mathbf{X}\|_F^2 \leq 3\epsilon\|\mathbf{PX}\|_F^2 + \epsilon d^2(\mathbf{P}, L_\mathbf{X})$$

which establishes the lemma. $\qquad\square$

**Theorem 13.** *Algorithm 4 outputs a subspace embedding with probability at least $1 - \delta$. In expectation Step 3 is run only a constant number of times with expected time $O(d^3 r^2/\epsilon^2)$.*

*Proof.* For each $j$, $\mathbf{H}_j \mathbf{A}$ succeeds with probability 99/100, meaning that for all $x$ we have $\|\mathbf{H}_j \mathbf{A}x\|_2 = (1 \pm \epsilon/9)\|\mathbf{A}x\|_2$. Suppose for some $j \neq j'$, $\mathbf{H}_j \mathbf{A}$ and $\mathbf{H}_{j'} \mathbf{A}$ are both successful. By definition we have

$$\|\mathbf{H}_j \mathbf{A}x\|_2 = (1 \pm \epsilon/3)\|\mathbf{H}_{j'} \mathbf{A}x\|_2$$

for all $x$. Taking the SVD of the embeddings, this is equivalent to

$$\|\mathbf{\Sigma}_j \mathbf{V}_j^\top x\|_2 = (1 \pm \epsilon/3)\|\mathbf{\Sigma}_{j'} \mathbf{V}_{j'}^\top x\|_2$$

---

**Algorithm 4** Boosting success probability of embedding

---

**Input:** $\mathbf{A} \in \mathbb{R}^{n \times d}$, parameters $\epsilon, \delta$.

1: Construct $r = O(\log \frac{1}{\delta})$ independent subspace embeddings $\mathbf{H}_j \mathbf{A}$, each having accuracy $\epsilon/9$ and success probability $99/100$.

2: Compute SVD $\mathbf{H}_j \mathbf{A} = \mathbf{U}_j \mathbf{\Sigma}_j \mathbf{V}_j^\top$ for $j \in [r]$.

3: **for** $j \in [r]$ **do**

4:    Check if for at least half $j' \neq j$,

$$\sigma_i(\mathbf{\Sigma}_{j'} \mathbf{V}_{j'}^\top \mathbf{V}_j \mathbf{\Sigma}_j^{-1}) \in [1 \pm \epsilon/3], \forall i.$$

5:    If so, output $\mathbf{H}_j \mathbf{A}$.

6: **end for**

---

---

**Algorithm 5** Randomized SVD [11]

---

**Input:** matrix $\mathbf{A} \in \mathbb{R}^{\ell \times d}$; parameters $t, q \in \mathbb{N}_+$.

1: ▷ Stage A

2: Generate an $\ell \times 2t$ Gaussian test matrix $\mathbf{\Omega}$.

3: Set $\mathbf{Y} = (\mathbf{A}^\top \mathbf{A})^q \mathbf{A}^\top \mathbf{\Omega}$, and compute QR-factorization: $\mathbf{Y} = \mathbf{Q}\mathbf{R}$.

4: ▷ Stage B

5: Set $\mathbf{B} = \mathbf{A}\mathbf{Q}$, and compute SVD: $\mathbf{B} = \mathbf{U}\mathbf{\Sigma}\tilde{\mathbf{V}}^\top$.

6: Set $\mathbf{V} = \mathbf{Q}\tilde{\mathbf{V}}$.

**Output:** $\mathbf{\Sigma}, \mathbf{V}$.

---

for all $x$. Making the change of variable $y := \mathbf{\Sigma}_j \mathbf{V}_j^\top x$, this is equivalent to

$$\|y\|_2 = (1 \pm \epsilon/3)\|\mathbf{\Sigma}_{j'} \mathbf{V}_{j'}^\top \mathbf{V}_j \mathbf{\Sigma}_j^{-1} y\|_2$$

for all $y$, which is true if and only if all singular values of $\mathbf{\Sigma}_{j'} \mathbf{V}_{j'}^\top \mathbf{V}_j \mathbf{\Sigma}_j^{-1}$ are in $[1 - \epsilon/3, 1 + \epsilon/3]$.

Conversely, if all singular values of $\mathbf{\Sigma}_{j'} \mathbf{V}_{j'}^\top \mathbf{V}_j \mathbf{\Sigma}_j^{-1}$ are in $[1 - \epsilon/3, 1 + \epsilon/3]$, one can trace the steps backward to conclude that $\|\mathbf{H}_j \mathbf{A}x\|_2 = (1 \pm \epsilon/3)\|\mathbf{H}_{j'} \mathbf{A}x\|_2$ for all $x$.

Since with probability at least $1 - \delta$, a $9/10$ fraction of the embeddings succeed with accuracy $\epsilon/9$, there exists a $j$ that can pass the test. It follows that any index $j$ which passes the test in the algorithm with a majority of the $j' \neq j$ is a successful subspace embedding with accuracy $\epsilon$.

Moreover, if we choose a random $j$ to compare to the remaining $j'$, the expected number of choices of $j$ until the test passes is only constant. Then finding the index $j$ only takes an expected $O(r)$ SVDs.

The time to do the SVD naively is $O(d^4/\epsilon^2)$. We can improve this by letting $\mathbf{T}$ be a fast Johnson-Lindenstrauss transform matrix of dimension $O(dr/\epsilon^2) \times O(d^2/\epsilon^2)$, then we can replace $\mathbf{H}_j \mathbf{A}$ with $\mathbf{T}\mathbf{H}_j \mathbf{A}$ for all $j \in [d]$. Then the verification procedure would only take $O(d^3 r^2/\epsilon^2)$ time. $\qquad\square$

### D.2 Proofs for Randomized SVD

The details of randomized SVD are presented in Algorithm 5, rephrased in our notations. We have the following analog of Lemma 1.

**Lemma 14.** *Let* $\mathbf{A} \in \mathbb{R}^{\ell \times d}$ *be an* $\ell \times d$ *matrix* $(\ell > d)$. *Let* $\epsilon \in (0, 1]$, $k, t \in \mathbb{N}_+$ *with* $d - 1 \geq t \geq k + \lceil 6k/\epsilon^2 \rceil - 1$. *Let* $\widehat{\mathbf{A}} = \mathbf{A}\mathbf{V}\mathbf{V}^\top$ *where* $\mathbf{V}$ *is computed by Algorithm 5 with* $q = O(\log \max\{\ell, d\})$. *Then with probability at least* $1 - 3e^{-t}$, *for any matrix* $\mathbf{X}$ *with* $d$ *rows and* $\|\mathbf{X}\|_F^2 \leq k$, *we have*

$$\|(\mathbf{A} - \widehat{\mathbf{A}})\mathbf{X}\|_F^2 \quad \leq \quad \frac{\epsilon^2}{3} \sum_{i=k+1}^d \sigma_i^2(\mathbf{A}),$$

$$\left| \|\mathbf{A}\mathbf{X}\|_F^2 - \|\widehat{\mathbf{A}}\mathbf{X}\|_F^2 \right| \quad \leq \quad \epsilon \sum_{i=k+1}^d \sigma_i^2(\mathbf{A}) + 2\epsilon\|\mathbf{A}\mathbf{X}\|_F^2.$$

*The algorithm runs in time $O(qt\ell d + t^2(\ell + d))$.*

*Proof.* As stated in Section 10.4 in [11], with probability at least $1 - 3e^{-t}$, we have

$$\|\mathbf{A} - \widehat{\mathbf{A}}\|_S \leq 2\sigma_{t+1}(\mathbf{A}). \tag{24}$$

Then we have

$$\|(\mathbf{A} - \widehat{\mathbf{A}})\mathbf{X}\|_F^2 \leq \|\mathbf{X}\|_F^2 \|\mathbf{A} - \widehat{\mathbf{A}}\|_S^2 \leq 2k\sigma_{t+1}^2(\mathbf{A})$$

where the first inequality follows because the spectral norm is consistent with the Euclidean norm, and the second inequality follows from (24). For our choice of $t$, we have

$$k\sigma_{t+1}^2(\mathbf{A}) \leq \frac{\epsilon^2}{6}(t - k + 1)\sigma_{t+1}^2(\mathbf{A}) \leq \frac{\epsilon^2}{6}\sum_{i=k+1}^{t+1}\sigma_i^2(\mathbf{A}) \leq \frac{\epsilon^2}{6}\sum_{i=k+1}^{d}\sigma_i^2(\mathbf{A}) \leq \frac{\epsilon^2}{6}d^2(\mathbf{A}, L_{\mathbf{X}}),$$

which leads to the first claim in the lemma.

To prove the second claim, first note that

$$\left|\|\mathbf{A}\mathbf{X}\|_F - \|\widehat{\mathbf{A}}\mathbf{X}\|_F\right|^2 \leq \|(\mathbf{A} - \widehat{\mathbf{A}})\mathbf{X}\|_F^2 \leq \frac{\epsilon^2}{3}d^2(\mathbf{A}, L_{\mathbf{X}}).$$

Then by Fact 1, we have

$$\left|\|\mathbf{A}\mathbf{X}\|_F^2 - \|\widehat{\mathbf{A}}\mathbf{X}\|_F^2\right| \leq \frac{3}{\epsilon}\left|\|\mathbf{A}\mathbf{X}\|_F - \|\widehat{\mathbf{A}}\mathbf{X}\|_F\right|^2 + 2\epsilon\|\mathbf{A}\mathbf{X}\|_F^2 \leq \epsilon d^2(\mathbf{A}, L_{\mathbf{X}}) + 2\epsilon\|\mathbf{A}\mathbf{X}\|_F^2$$

which completes the proof. $\qquad\square$

### D.3 Proof of Theorem 6

Let $\mathbf{T}$ to be a diagonal block matrix with $\mathbf{H}_1, \mathbf{H}_2, \ldots, \mathbf{H}_s$ on the diagonal. Then Algorithm 2 is just to run Algorithm disPCA on $\mathbf{TP}$ to get the principal components $\mathbf{V}$. Recall that the goal is to show $\tilde{\mathbf{P}} = \mathbf{P}\mathbf{V}\mathbf{V}^\top$ is a good proxy for the original data $\mathbf{P}$ with respect to $\ell_2$ error fitting problems. It suffices to show that $\tilde{\mathbf{P}}$ satisfies enjoys properties similar to those stated in Lemma 4.

To prove this, we begin with a lemma saying that $\mathbf{T}\tilde{\mathbf{P}}$ enjoys such properties, i.e. such properties are approximately preserved when replacing exact SVD with randomized SVD in Algorithm disPCA (Lemma 15). Then we can show that $\tilde{\mathbf{P}}$ enjoys similar properties as $\mathbf{T}\tilde{\mathbf{P}}$, i.e. these properties are approximately preserved under subspace embedding (Lemma 17).

**Lemma 15.** *For any $d \times k$ matrix $\mathbf{X}$ with orthonormal columns,*

$$\|\mathbf{TPX} - \mathbf{T}\tilde{\mathbf{P}}\mathbf{X}\|_F^2 \leq O(\epsilon^2)d^2(\mathbf{TP}, L_{\mathbf{X}}) + O(\epsilon^3)\|\mathbf{TPX}\|_F^2,$$

$$\left|\|\mathbf{TPX}\|_F^2 - \|\mathbf{T}\tilde{\mathbf{P}}\mathbf{X}\|_F^2\right| \leq O(\epsilon)d^2(\mathbf{TP}, L_{\mathbf{X}}) + O(\epsilon)\|\mathbf{TPX}\|_F^2.$$

*Proof.* The proof follows that of Lemma 4 to $\mathbf{TP}$. But now exact SVD is replaced with randomized SVD, so we need to argue that randomized SVD produces similar result as exact SVD in the sense of Lemma 7. This is already proved in Lemma 14. Also note that we need a technical lemma bounding the small error terms incurred on the intermediate result $\mathbf{T}\widehat{\mathbf{P}}$. This is done by Lemma 16. $\qquad\square$

**Lemma 16.**

$$\|\mathbf{T}\widehat{\mathbf{P}}\mathbf{X}\|_F^2 \leq \epsilon d^2(\mathbf{TP}, L_{\mathbf{X}}) + (1 + 2\epsilon)\|\mathbf{TPX}\|_F^2,$$

$$d^2(\mathbf{T}\widehat{\mathbf{P}}, L_{\mathbf{X}}) \leq (1 + \epsilon)d^2(\mathbf{TP}, L_{\mathbf{X}}) + \epsilon\|\mathbf{TPX}\|_F^2.$$

*Proof.* For the first statement, by Lemma 14, we have

$$
\begin{aligned}
\left|\|\mathbf{T}\widehat{\mathbf{P}}\mathbf{X}\|_F^2 - \|\mathbf{TPX}\|_F^2\right| &\leq \sum_{i=1}^{s}\left|\|\mathbf{TP}_i\mathbf{X}\|_F^2 - \|\mathbf{T}\widehat{\mathbf{P}}_i\mathbf{X}\|_F^2\right| \\
&\leq \epsilon\sum_{i=1}^{s}d^2(\mathbf{TP}_i, L_{\mathbf{X}}) + 2\epsilon\sum_{i=1}^{s}\|\mathbf{TP}_i\mathbf{X}\|_F^2 \\
&\leq \epsilon d^2(\mathbf{TP}, L_{\mathbf{X}}) + 2\epsilon\|\mathbf{TPX}\|_F^2. \tag{25}
\end{aligned}
$$

For the second statement, by Pythagorean Theorem,

$$
\begin{aligned}
d^2(\mathbf{T}\widehat{\mathbf{P}}, L_{\mathbf{X}}) - d^2(\mathbf{T}\mathbf{P}, L_{\mathbf{X}}) &= \left[ \|\mathbf{T}\widehat{\mathbf{P}}\|_F^2 - \|\mathbf{T}\widehat{\mathbf{P}}\mathbf{X}\|_F^2 \right] - \left[ \|\mathbf{T}\mathbf{P}\|_F^2 - \|\mathbf{T}\mathbf{P}\mathbf{X}\|_F^2 \right] \\
&= \left[ \|\mathbf{T}\widehat{\mathbf{P}}\|_F^2 - \|\mathbf{T}\mathbf{P}\|_F^2 \right] + \left[ \|\mathbf{T}\mathbf{P}\mathbf{X}\|_F^2 - \|\mathbf{T}\widehat{\mathbf{P}}\mathbf{X}\|_F^2 \right] \\
&\leq \|\mathbf{T}\mathbf{P}\mathbf{X}\|_F^2 - \|\mathbf{T}\widehat{\mathbf{P}}\mathbf{X}\|_F^2.
\end{aligned}
$$

The second statement then follows from the last inequality and (25). $\qquad \square$

**Lemma 17.** *For any $d \times k$ matrix $\mathbf{X}$ with orthonormal columns,*

$$
\begin{aligned}
\|\mathbf{P}\mathbf{X} - \tilde{\mathbf{P}}\mathbf{X}\|_F^2 &\leq O(\epsilon^2)d^2(\mathbf{P}, L_{\mathbf{X}}) + O(\epsilon^3)\|\mathbf{P}\mathbf{X}\|_F^2, \\
\left| \|\mathbf{P}\mathbf{X}\|_F^2 - \|\tilde{\mathbf{P}}\mathbf{X}\|_F^2 \right| &\leq O(\epsilon)d^2(\mathbf{P}, L_{\mathbf{X}}) + O(\epsilon)\|\mathbf{P}\mathbf{X}\|_F^2.
\end{aligned}
$$

*Proof.* By the property of subspace embedding, we have $\|\mathbf{T}\mathbf{P}\mathbf{X} - \mathbf{T}\tilde{\mathbf{P}}\mathbf{X}\|_F^2 = (1 \pm \epsilon)\|\mathbf{P}\mathbf{X} - \tilde{\mathbf{P}}\mathbf{X}\|_F^2$, $\|\mathbf{T}\mathbf{P}\mathbf{X}\|_F^2 = (1 \pm \epsilon)\|\mathbf{P}\mathbf{X}\|_F^2$ and $d^2(\mathbf{T}\mathbf{P}, L_{\mathbf{X}}) = \|\mathbf{T}\mathbf{P} - \mathbf{T}\mathbf{P}\mathbf{X}\mathbf{X}^\top\|_F^2 = (1 \pm \epsilon)\|\mathbf{P} - \mathbf{P}\mathbf{X}\mathbf{X}^\top\|_F^2 = (1 \pm \epsilon)d^2(\mathbf{P}, L_{\mathbf{X}})$. Then

$$
\begin{aligned}
(1 + \epsilon)\|\mathbf{P}\mathbf{X} - \tilde{\mathbf{P}}\mathbf{X}\|_F^2 &\leq \|\mathbf{T}\mathbf{P}\mathbf{X} - \mathbf{T}\tilde{\mathbf{P}}\mathbf{X}\|_F^2 \\
&\leq O(\epsilon^2)d^2(\mathbf{T}\mathbf{P}, L_{\mathbf{X}}) + O(\epsilon^3)\|\mathbf{T}\mathbf{P}\mathbf{X}\|_F^2 \\
&\leq O(\epsilon^2)d^2(\mathbf{P}, L_{\mathbf{X}}) + O(\epsilon^3)\|\mathbf{P}\mathbf{X}\|_F^2
\end{aligned}
$$

where the second inequality is from Lemma 15. This then leads to the first statement.

For the second statement, we have

$$
\begin{aligned}
(1 + \epsilon)\|\mathbf{P}\mathbf{X}\|_F^2 - (1 - \epsilon)\|\tilde{\mathbf{P}}\mathbf{X}\|_F^2 &\leq \|\mathbf{T}\mathbf{P}\mathbf{X}\|_F^2 - \|\mathbf{T}\tilde{\mathbf{P}}\mathbf{X}\|_F^2 \\
&\leq O(\epsilon)d^2(\mathbf{T}\mathbf{P}, L_{\mathbf{X}}) + O(\epsilon)\|\mathbf{T}\mathbf{P}\mathbf{X}\|_F^2 \\
&\leq O(\epsilon)d^2(\mathbf{P}, L_{\mathbf{X}}) + O(\epsilon)\|\mathbf{P}\mathbf{X}\|_F^2
\end{aligned}
$$

which leads to

$$
\|\mathbf{P}\mathbf{X}\|_F^2 - \|\tilde{\mathbf{P}}\mathbf{X}\|_F^2 \leq O(\epsilon)d^2(\mathbf{P}, L_{\mathbf{X}}) + O(\epsilon)\|\mathbf{P}\mathbf{X}\|_F^2.
$$

A similar argument bounds $\|\tilde{\mathbf{P}}\mathbf{X}\|_F^2 - \|\mathbf{P}\mathbf{X}\|_F^2$, which completes the proof. $\qquad \square$

We represent Theorem 6 in a general form for $\ell_2$-error geometric fitting problems.

**Theorem 6.** *Suppose Algorithm 2 takes $\epsilon \in (0, 1/2], t_1 = t_2 = O(\max\left\{ \frac{k}{\epsilon^2}, \log \frac{s}{\delta} \right\}), \ell = O(\frac{d^2}{\epsilon^2}), q = O(\max\{\log \frac{d}{\epsilon}, \log \frac{sk}{\epsilon}\})$ as input, and sets the failure probability of each local subspace embedding to $\delta' = \delta/2s$. Let $\tilde{\mathbf{P}} = \mathbf{P}\mathbf{V}\mathbf{V}^\top$. Then with probability at least $1 - \delta$, there exists a constant $c_0 \geq 0$, such that for any set of $k$ points $\mathcal{L}$,*

$$
(1 - \epsilon)d^2(\mathbf{P}, \mathcal{L}) - \epsilon\|\mathbf{P}\mathbf{X}\|_F^2 \leq d^2(\tilde{\mathbf{P}}, \mathcal{L}) + c_0 \leq (1 + \epsilon)d^2(\mathbf{P}, \mathcal{L}) + \epsilon\|\mathbf{P}\mathbf{X}\|_F^2
$$

*where $\mathbf{X}$ is an orthonormal matrix whose columns span $\mathcal{L}$. The total communication is $O(skd/\epsilon^2)$ and the total time is $O\left(\mathrm{nnz}(\mathbf{P}) + s\left[\frac{d^3k}{\epsilon^4} + \frac{k^2d^2}{\epsilon^6}\right] \log \frac{d}{\epsilon} \log \frac{sk}{\delta\epsilon}\right)$.*

*Proof.* The proof of correctness follows the proof of Theorem 3, replacing the use of Lemma 4 with Lemma 17.

On each node $v_i$, the subspace embedding takes time $O(\mathrm{nnz}(\mathbf{P}_i))$, and the randomized SVD takes time $O(qt_1\ell d + t_1^2(\ell + d))$; on the central coordinator, the randomized SVD takes time $O(qt_1(st_1)d + t_1^2(st_1 + d))$ since $\mathbf{Y}$ has $O(st_1)$ non-zero rows. The total running time then follows from the choice of the parameters. The total communication cost follows from the fact that the algorithm only sends $\mathbf{\Sigma}_i^{(t_1)}, \mathbf{V}_i^{(t_1)}$ from each node to the central coordinator. $\qquad \square$