[Reviews · NeurIPS 2014]

Submitted by Assigned_Reviewer_3

This paper considers the problem of trading off the communication and computation cost of distributed computation and proposes a new distributed k L-2 error fitting algorithm. The proposed algorithm can be seen as a combination of many previous speed up techniques for distributed PCA and clustering methods. However, the authors also contribute optimizations over the base methods and further improves the communication and computation efficiency. The theoretical guarantee is sound and experiments are convincing.

What is the subroutine of A_\alpha in Algorithm 1? Moreover, algorithm 2 involves algorithm 1 which uses A_\alpha. However, it seems in Theorem 6 that both the communication and computation cost is independent of \alpha? Could the authors provide some explanations?

Line 275, how shall we understand that for the subspace embedding, there is \|HAy\|_2 = (1\pm \epsilon)\|Ay\|_2?

In Theorem 6, the failure probability depends on s and t. I suppose t is equal to t1 and t2, which also depends on s. Could the authors optimize the probability and remove the dependence on t? As for the current statement, suppose t1=t2=O(\log s), then the probability is 1-O(1/s+1). Is this result good enough?

Could the authors provide the explicit form for the constant c_0? Such that the readers can better understand the tightness of the bound.

My biggest concern is about the performance measure. As mentioned in line 85, it is desired to find a center set \mathcal{L}’ such that the relative error is small, d^2(P, \mathcal{L}’ ) \leq (1+\epsilon) \min_{\mathcal{L}}d^2(P,\mathcal{L}). It is expected to see a similar relative error bound for algorithm 2 in Theorem 6.

In the experiments, the authors should provide the value of parameter \epsilon, or the values of t1 and t2.

Minor comments:
Line 93, I suggest to use “reducing communication cost” as the paragraph title, which more clearly expressed the contributions compared with “improved communication”. Similarly, line 108, “improved computation” should also be revised.
Summary: Though the novelty of this paper is not so significant, it provides an insightful analysis on the communication and computation trade-off for distributed algorithms. Some details were not clear, but authors clarify them in the rebuttal well.

Submitted by Assigned_Reviewer_8

The paper studies principal component analysis (PCA) in a distributed setting. The paper presents new algorithms and analyses for distributed PCA with reduced communication and computation cost. A good solid paper overall; the writing is clear, novelty and significance are high. One minor comment regarding the literature survey: looking at the computational cost of SVD on each cluster (page 3, first para) , it is not clear to me how the authors claim the cost to be min(n_i d^2, n_i^2 d). There have been several memory efficient and computationally cheaper algorithms for PCA proposed recently; for instance see the “stochastic optimization for PCA with capped MSG” at last year’s NIPS. I believe the cost to be linear in d and overall runtime to be O(dk^2/eps^2) for an eps-suboptimal solution. Actually there has been a surge of interest in scalable algorithms for PCA; the related work section would benefit from that survey (look for PCA papers at last year’s NIPS).

====

Regarding authors' response about MSG for PCA, my understanding is

(a) it is straightforward to give guarantees in the online setting, in fact MEG, which is an alternative to MSG (both are instances of mirror descent with different potential functions), was first studied by Warmuth and Kuzmin in the online setting; an earlier paper "Stochastic optimization for PCA and PLS" by the same authors makes the connection clearer,

(b) the capped version suggests a cap of k+1 on the overall rank which makes the problem non-convex but is still tractable; this variant enjoys a computational cost of O(k^3)
Summary: The paper studies principal component analysis (PCA) in a distributed setting. A good paper.

Submitted by Assigned_Reviewer_43

This paper suggests a distributed PCA and k-means algorithms and more importantly rigorously proves competitive communication cost and computational efficiency for a given accuracy for these algorithms (and even a generalized set of problems). The paper uses various methods to modify and improve the communication and computation of previous methods and the emphasis is on the theory supporting it. It improves the communication by first projecting the data on a lower-dimensional subspace via initial approximate distributed PCA (following [9]) and then running existing algorithms in the reduced space. It improves the computation by oblivious subspace embedding.

The paper is limited to the case of a central processor, which has to communicate with all processors (but its memory may not be shared). As far as I understand it is a standard assumption in analysis of distributed algorithms and even with this assumption the theoretical contribution is important.

Overall the paper is well written, though beyond the introduction it takes some time to carefully understand it.

I read the other reviews and the rebuttal. I did not change the text of the above review and the quality score. However, I have changed my mind regarding the impact score. As I mentioned earlier I have no direct expertise in distributed algorithms and I was not familiar with many of the cited works. It is thus hard for me to truly judge the impact of this work on the area. I find it interesting and valuable for a broad audience in machine learning. However, since it seems to combine some previous ideas and has no real surprise (though still interesting), I believe its impact score is 1 and not 2.
Summary: This is an interesting theoretical paper on verified improved communication cost and computational efficiency of an algorithm for distributed PCA (and other related algorithms).
Author Feedback
Author rebuttal: We thank the reviewer for valuable comments.

Reviewer#1
**Algorithm 1
A_\alpha can be any non-distributed algorithm that outputs an alpha-approximation for k-means (see the first line in Algorithm 1). For example, it can be the local search algorithm in the paper "a local search approximation algorithm for k-means clustering" by Kanungo, Tapas, et al., which achieves a constant approximation factor.
Our algorithm calls the distributed k-means clustering algorithm in [3], which then uses A_\alpha as a subroutine. However, A_\alpha is non-distributed (as noted in the first line in Algorithm 1), and it has no contribution to the communication.

**Line 275
When y runs over all vectors in R^d, Ay produces a subspace in R^n. The linear mapping H approximately preserves the l_2 norm of the vectors in this subspace.

**The failure probability in Theorem 6
Yes, t is equal to t1 and t2. Here for ease of presentation, we aim at a constant success probability. Typically, we deal with s=100, and we can choose t> 20\log s, so that the success probability can be 0.9. Furthermore, to achieve an arbitrary failure probability \delta, we can simply set t=\log (s/\delta) and set the failure probability of each subspace embedding to be \delta/2s.

**The explicit form for the constant c_0 in Theorem 6
c_0 = \| P \|_F - \| \tilde{P} \|_F, that is, the difference of the Frobenius norm of the original data matrix and the projected data matrix. We do not provide an explicit form of c_0, since it does not affect the final approximation bound. More precisely, as pointed out in Line 224-227, the guarantee of Theorem 3 and Theorem 6 implies that any \alpha-approximation for the projected data is a (1+3\epsilon)\alpha-approximation for the original data. This approximation bound holds for any value of c_0.

**Performance measure in Theorem 6
As pointed out in Line 224-227, the guarantee of Theorem 3 and Theorem 6 implies that any \alpha-approximation for the projected data is a (1+3\epsilon)\alpha-approximation for the original data. That is, the disPCA step introduces a small (1+3\epsilon) multiplicative error. Due to space limitation, we only give a concrete application of Theorem 3 on k-means clustering in Theorem 5. But the same relative error bound can be achieved from Theorem 6 using the same argument.

**Parameter values in the experiments
We will add the descriptions of these parameter values in our later version.

Reviewer#3
We thank the reviewer for pointing to the literature. We will include these related works in our later version.

Regarding the memory efficient and computationally cheap algorithms for PCA, such as those proposed in "Stochastic Optimization for PCA with Capped MSG", we thank the reviewer for pointing them out and we can certainly test their empirical performance. We would like to mention though that regarding the claimed cost of min(n_i d^2, n_i^2 d), this was claimed in the context of worst-case SVD running time complexity. For the capped MSG PCA algorithms proposed above, we first want to point out that they assume an underlying distribution on the rows of the n x d matrix with a certain 4-th moment condition. Also, the running time can be d^3 in the worst-case. See Section 2 and Section 4 of
http://arxiv.org/pdf/1307.1674v1.pdf
regarding these two claims.